# Optimal Decisions in a Sea-Cargo Supply Chain with Two Competing Freight Forwarders Considering Altruistic Preference and Brand Investment

Xiao-Ying Ma [1], Duo-Qing Sun [1,2,*], Shu-Xia Liu [3], Yue-Ting Li [1], Hui-Quan Ma [1], Ling-Min Zhang [1] and Xia Li [1]

1   Institute of Mathematics and Systems Science, Hebei Normal University of Science and Technology, Qinhuangdao 066004, China; maxy6688@163.com (X.-Y.M.); 18502206853@163.com (Y.-T.L.); mhqlyz@163.com (H.-Q.M.); lingmin9999@163.com (L.-M.Z.); lixia_snow@163.com (X.L.)
2   Hefei Innovation Research Institute, Beihang University, Hefei 230012, China
3   HEKRI of Marine Economy and Coastal Economic Zone, Hebei Normal University of Science and Technology, Qinhuangdao 066004, China; xxglliushuxia@126.com
*   Correspondence: sun_duoqing@126.com

**Abstract:** Maritime transportation is a crucial component of international cargo transport, offering several advantages, such as route flexibility, large capacity, and cost-effectiveness. The competition and collaboration among the node enterprises in the sea-cargo supply chain system (SCSCS) directly impact the overall structure and efficiency of the supply chain system, introducing complexity in analysis. This research focuses on a two-level SCSCS comprising one shipping company and two competing freight forwarders, considering their altruistic preferences manifested through contributing to the shipping company's brand building. Employing a Stackelberg game model, this study examines the effects of the shipping company's brand investment willingness and the freight forwarders' altruistic preferences on the decision making and profits of all stakeholders. The findings reveal that a higher willingness of the shipping company to invest in its brand building leads to increased profits for all parties involved. However, while the altruistic behaviors of the freight forwarders can enhance the shipping company's profits, their own profits may not necessarily see the same impact. Furthermore, moderate competition between the freight forwarders can enhance the profits for all members. This research identifies the circumstances in which the freight forwarders' altruistic preferences can lead to increased profits for themselves, achieving both altruistic and self-interested outcomes.

**Keywords:** sea-cargo supply chain; altruistic preference; brand investment; optimal decision; competition for freight forwarders



## 1. Introduction

Maritime transport plays a significant role in international freight transportation due to its advantages, such as flexible route selection and high carrying capacity [1,2]. In a complex sea-cargo supply chain system (SCSCS), shipping companies specialize in transportation and ship maintenance, while freight forwarders directly interact and provide services, including cargo packaging, booking cargo space, selecting routes, and insurance, to the shippers. The interaction between shipping companies and freight forwarders directly influences the overall structure and efficiency of the SCSCS, and balancing cooperation and competition is crucial for the stability and success of the SCSCS.

To enhance competitiveness and attract more orders, shipping companies invest in their brands by improving service quality and implementing marketing promotions, and by fostering a positive reputation within the industry. As a result, the improved brand value of shipping companies leads to an increase in their market share and overall revenues within the SCSCS [3,4]. It is important to note that the freight forwarding industry is

highly competitive. Some shipping companies have implemented marketing integration. For example, COSCO SHIPPING, China SHIPPING, and other large shipping enterprises have set up their own freight forwarders, thus undoubtedly reducing the living space of traditional freight forwarders with price difference and booking fees for profit purposes. Therefore, in order to maintain the stability of cooperation between the two sides in the process of long-term brand building and strengthen their position and profitability, the freight forwarder companies inevitably have altruistic behavior by caring about the profits of shipping companies while paying attention to their own profits. One way in which freight forwarders contribute to the success of shipping companies is by actively participating in brand-building efforts. They make regional investments and extend the reach of shipping companies' brands through their own operations. For example, Intent Logistics Co., Ltd. not only promotes itself but also contribute to enhancing the brand reputation of shipping companies (Accessed from http://www.szycil.com/show-15-18844 -1.html (accessed on 26 June 2023) or see Supplementary Material S1). This is because smart shippers need to consider various factors when choosing a freight forwarder, one of which is the reliability of transportation, that is, the shipping company entrusted by the freight forwarder should have the strength and credibility to reduce the possibility of becoming a victim of maritime fraud. These factors motivate our research to explore the influence of the altruistic preferences of competing freight forwarders on the decision-making process and profits within a sea-cargo supply chain system that consider brand investment.

Traditional supply chain management has been centered around the pursuit of profits and competitiveness, assuming that decision-makers are fully rational [5]. However, the emergence of behavioral operation management theory has questioned this assumption and highlighted the influence of decision-makers' behavioral perspectives on supply chains [6,7]. Various social preference behaviors, such as altruism, that are considered irrational factors in decision-making, play a significant role in shaping the overall decision-making process of supply chains.

Numerous enterprises exhibit altruistic preferences in their supply chain alliances to foster the system coordination [8–12]. For example, Apple adopts a practice of prepaying suppliers to ensure a steady supply of parts and maintain stable production capacity [10]. Toyota offers technical and management support to its suppliers to enhance productivity, while General Motors assists suppliers in upgrading their technologies [11]. The incorporation of altruistic preferences introduces intricacy in analyzing and managing supply chain systems, as it requires a deeper understanding of the motivations and behaviors of the participants beyond mere self-interest [10,11,13].

The exploration of supply chain systems incorporating altruistic preferences began with a controlled experiment conducted by Loch and Wu [14], who introduced an altruistic preference utility function, laying the groundwork for studying altruistic preferences. Following this, subsequent studies have employed this utility function to examine the effects of altruistic preferences on decision making, profits, utility of members, and the channel efficiency in one-to-one supply chain systems with price-dependent demand.

The literature relevant to our study focuses on the influence of altruistic preferences of supply chain members on decision making within the framework of game theory. For example, Ge and Hu [15] argued that although altruistic behaviors among the dominant players in a supply chain system can enhance the overall performance to some extent, centralized decision making achieves even higher performance. Shi et al. [16] discovered that the altruistic preferences of both manufacturers and retailers significantly affect pricing strategies. Wang et al. [17] revealed that altruistic preferences boost the revenue of the e-commerce platform but are detrimental to the remanufacturer. However, these studies did not analyze the range of altruistic preference coefficients to maximize profits. Practical cases and theoretical studies have shown that excessive altruism may benefit others while disadvantaging oneself and may even reduce the overall supply chain efficiency. A notable example is Amazon's development of AWS cloud services, which incurred substantial financial losses due to their altruistic behavior of serving various enterprise users. Therefore,

the introduction of altruistic behavior must be moderate. Sun et al. [18] showed through numerical analysis that excessive altruism of sellers would reduce the overall profit of a fresh agricultural product supply chain system. Some scholars theoretically provided value ranges for altruistic preference coefficients that can either benefit or harm supply chain members [19–25]. However, these studies do not consider competition among supply chain members.

Our work is also related to previous research on brand investment in SCSCS. Since the shippers have limited information available, the brand effect of the shipping companies will be taken into account when they choose shipping services. Consequently, the decision-making problem in supply chains with brand-dependent demand has garnered attention. For instance, Liu et al. [3] investigated the heterogeneous cooperation in brand investment between SCSCS members and found that cooperation between a shipping company and a freight forwarder enhances the overall SCSCS profit. Further, Zhu et al. [4] investigated branding inputs in an SCSCS dominated by shipping companies and concluded that a shipping company's brand investment and subsequent improvements in brand value benefit the freight forwarder's profit. However, these studies do not consider the altruistic preferences of supply chain members.

Another stream of literature closely related to our study focuses on optimal decision-making in SCSCS. However, previous studies in this field have not simultaneously considered both altruistic preferences and brand investments of supply chain members. To the best of our knowledge, Li et al. [26] is the only one that specifically examined brand investment. They investigated how the shipping company's brand investment willingness and the freight forwarder's altruistic preference affect the decisions and profits of the two members in a two-echelon SCSCS. However, their study only involved a single forwarder and did not account for competition among multiple forwarders. In reality, downstream firms in supply chains typically operate in a competitive environment [27]. Competition can have a significant impact on the system dynamics, pricing strategies, profitability, and collaborative opportunities available to the members.

Therefore, our study aims to fill these gaps by examining the optimal decision-making problem in a two-echelon supply chain consisting of one shipping company and two competing freight forwarders, both of whom possess altruistic preferences. In a game-theoretical framework, we establish two decision-making models, and by solving the models, try to give the optimal decisions in an SCSCS. This study intends to reveal the influence of altruistic preference on the optimal decisions of supply chain members in a competitive environment. This research provides valuable insights for decision-makers in the supply chain system, offering strategies to effectively navigate the competitive environment and gain a competitive advantage.

The literature most relevant to our study is summarized in Table 1 to show the innovation and contribution of this paper.

**Table 1.** Comparison of the related literature.

| Literature | Altruistic Preference | Brand Investment | Competition | SCSCS |
|:---:|:---:|:---:|:---:|:---:|
| [3,4] | | √ | | √ |
| [10,14–25] | √ | | | |
| [26] | √ | √ | | √ |
| Our study | √ | √ | √ | √ |

The contributions of this study can be summarized as follows. First, it is the first investigation, to the best of our knowledge, that explores an SCSCS involving two competing freight forwarders with altruistic preferences. This study develops two Stackelberg game models to examine the collaborative dynamics between the dominant shipping company, responsible for brand investment, and its followers, the freight forwarders, who extend the shipping company's brand value through their investments. Second, the study presents

optimal solutions for both models, with one incorporating the altruistic preferences of the freight forwarders and the other without considering them. Third, this study analyzes how altruistic preferences and the willingness in the brand influence the optimal decisions and profits of the supply chain members in a competitive environment. Lastly, it should be emphasized that existing literature on altruistic preferences does not consider competition among supply chain members. Thus, our research results also contribute to the literature on optimal decisions in supply chain systems with altruistic and competitive members, not only limited to the SCSCS.

The remainder of this paper proceeds as follows. Section 2 describes the problem and constructs basic models. Section 3 provides the equilibrium decisions of the supply chain members in the two cases of the absence of altruistic preference and the presence of the altruistic preference of each forwarder, respectively. Section 4 conducts parametric analysis on the equilibrium results. Section 5 presents numerical analyses. Section 6 includes some discussion and conclusions. It highlights the main conclusions of this study and gives directions for future research. All proofs of lemmas and propositions are presented in Supplementary Materials.

## 2. Problem Description and Basic Model

This study considers a two-echelon SCSCS comprising one shipping company and two competing freight forwarders in the shipping market. The shipping company offers freight services to shippers through these freight forwarders. The interactions between the shipping company and the freight forwarders follow a Stackelberg game, with the shipping company assuming the dominant role.

Both the shipping company and the freight forwarders are motivated to enhance their market share and competitiveness. They actively participate in the shipping company's brand-building activities within the context of the collaborative supply chain. The shipping company focuses on improving service quality, upgrading management techniques, and reengineering operational processes. Simultaneously, the freight forwarders contribute to the brand-building endeavors of the shipping company. They invest in regional advertising, marketing promotions, and other strategies to boost its short-term performance, which aligns with and complements the long-term brand building undertaken by the shipping company.

In the subsequent discussion, the unit shipping prices charged by the shipping company to freight Forwarders 1 and 2 are denoted as $w_1$ and $w_2$, respectively, and are related to the brand value of the shipping company, $e(>0)$. We will demonstrate in Section 3 that $w_1 = w_2$ for the equilibrium decisions. The corresponding unit freight service prices charged by freight Forwarders 1 and 2 to the shippers, based on the above unit shipping prices, are denoted as $p_1$ and $p_2$, respectively. The market demand ($Q_i$, $i =$1, 2) of each freight forwarder is described by the following linear functions:

$$Q_1(p_1, p_2, e, t_1) = k - p_1 + \mu p_2 + \lambda e + \eta t_1 \tag{1}$$

$$Q_2(p_1, p_2, e, t_2) = k - p_2 + \mu p_1 + \lambda e + \eta t_2 \tag{2}$$

These linear demand functions are widely used in economics and supply chain management research [28–33]. Here, the parameter $k(>0)$ represents the potential market size. Without loss of generality, the price-sensitive coefficients are normalized to 1 [34–38]. The competition coefficient between the two freight Forwarders 1 and 2, $\mu$, satisfies $0 < \mu < 1$ [39–41], where a larger value indicates higher competition intensity. It should be emphasized that $0 < \mu < 1$ is used to acknowledge the greater influence of each forwarder's own pricing on its market demand compared to the pricing of its competitor. The brand preference of the shippers, denoted by $\lambda$, measures the sensitivity of market demand to the brand value. $t_1$ and $t_2$ represent the effort levels of freight Forwarders 1 and 2, respectively, in extending the shipping company's brand value, $e$. The sensitivity of market demand to the above effort levels is represented by $\eta(>0)$.

The expenses of the shipping company consist of two parts: (1) the marginal cost ($c$) for providing shipping services, and (2) the fixed investment cost ($I_s$) for enhancing brand value. $I_s$ is a convex function with respect to the brand value, and it is assumed that $I_s = \alpha e^2$ [26], where $\alpha$(>0) is the coefficient of the shipping company's fixed investment cost for brand value improvement. A higher value of $\alpha$ implies more challenges in improving brand value.

According to [42,43], the effort costs for freight Forwarders 1 and 2 to extend brand value are $I_{f_1} = \beta t_1^2$ and $I_{f_2} = \beta t_2^2$, respectively. Note that the quadratic functions capture two important features of the effort costs [42]. First, they are strictly increasing, which is consistent with the fact that higher levels of efforts come with higher costs. Second, the convexity of the functions suggests that exerting more efforts actually yield diminishing marginal returns. $\beta$(>0) is the cost sensitivity coefficient of the effort.

For clarity, we summarize the notations in Table 2.

**Table 2.** Notations.

| Notations | Descriptions |
|---|---|
| Index | |
| $i$ | Two competing freight forwarders, $i \in \{1, 2\}$ |
| Decision variables | |
| $w_i$ | Unit shipping price charged by the shipping company to freight forwarder $i$ |
| $e$ | Brand value of the shipping company |
| $p_i$ | Unit freight service prices charged by freight forwarder $i$ to the shippers |
| $t_i$ | Effort level of freight forwarder $i$ |
| Parameters | |
| $c$ | Shipping company's marginal cost for providing shipping services |
| $k$ | Potential market size |
| $\mu$ | Competition coefficient between the two freight forwarders |
| $\lambda$ | Brand preference of the shippers |
| $\eta$ | Sensitivity of market demand to the effort level of freight forwarders |
| $\alpha$ | Coefficient of the shipping company's fixed investment cost for brand value improvement |
| $\beta$ | Effort cost sensitivity coefficient |

More notations will be described later when needed.

Next, we propose several assumptions underlying the proposed model.

**Assumption 1.** *There is a maximum limit of the impacts of effort levels for freight forwarders on demands because market demands are limited. We assume that $A = \eta^2 / 4\beta < 1$. This means that $\eta < \sqrt{4\beta}$. The mathematical meaning of this assumption can be seen from the subsequent mathematical derivation.*

Because we focus on positive prices, effort levels, market demands, and profits at equilibrium, we need stricter restrictions. For this, let

$$B = 2A = \eta B_0, \ B_0 = \eta / 2\beta \tag{3}$$

and make the following assumptions.

**Assumption 2.** *$B < 2 - \mu$, this is, $A < 1 - \mu/2$ or $\eta < \sqrt{2\beta(2 - \mu)}$.*

It should be mentioned that Assumption 1 also holds under Assumption 2, and from Assumption 2, it is easy to see that

$$2 - B + \mu > 0 \tag{4}$$

$$(-2 + B)^2 - \mu^2 = (2 - B - \mu)(2 - B + \mu) > 0 \tag{5}$$

**Assumption 3.** *The potential market scale is relatively large. Specifically, we assume that the potential market scale satisfies $k > c(1 - \mu)$.*

In addition, it is assumed that the shipping company and freight forwarders are risk-neutral and that information is symmetric among all parties involved.

Based on these assumptions, the profit functions of the shipping company, freight Forwarders 1 and 2, and the SCSCS can be expressed as follows:

$$\begin{aligned}\pi_s(w_1, w_2, e) &= (w_1 - c)Q_1 + (w_2 - c)Q_2 - \alpha e^2 \\ &= (w_1 - c)(k - p_1 + \mu p_2 + \lambda e + \eta t_1) + (w_2 - c)(k - p_2 + \mu p_1 + \lambda e + \eta t_2) - \alpha e^2\end{aligned} \tag{6}$$

$$\pi_{f_1}(p_1, t_1) = (p_1 - w_1)Q_1 - \beta t_1^2 = (p_1 - w_1)(k - p_1 + \mu p_2 + \lambda e + \eta t_1) - \beta t_1^2 \tag{7}$$

$$\pi_{f_2}(p_2, t_2) = (p_2 - w_2)Q_2 - \beta t_2^2 = (p_2 - w_2)(k - p_2 + \mu p_1 + \lambda e + \eta t_2) - \beta t_2^2 \tag{8}$$

$$\begin{aligned}\pi(p_1, p_2, e, t_1, t_2) &= (p_1 - c)(k - p_1 + \mu p_2 + \lambda e + \eta t_1) \\ &+ (p_2 - c)(k - p_2 + \mu p_1 + \lambda e + \eta t_2) - \alpha e^2 - \beta t_2^2 - \beta t_1^2\end{aligned} \tag{9}$$

## 3. Decisions

### 3.1. Decisions in the Absence of Altruistic Preference

In this section, the study focuses on determining the equilibrium solutions of all parties' decisions when the two freight forwarders are completely self-interested. The Stackelberg game between the shipping company and freight forwarders is carried out in two stages. (1) The shipping company, acting as the leader, prioritizes the shipping prices $(w_1, w_2)$ and the brand value $(e)$, and (2) according to the decision making of the shipping company, the two freight forwarders as followers decide their own effort levels for extending the brand value $(t_1, t_2)$ and the freight service prices $(p_1, p_2)$. The goal of all parties is to maximize their profits, as illustrated in Figure 1.

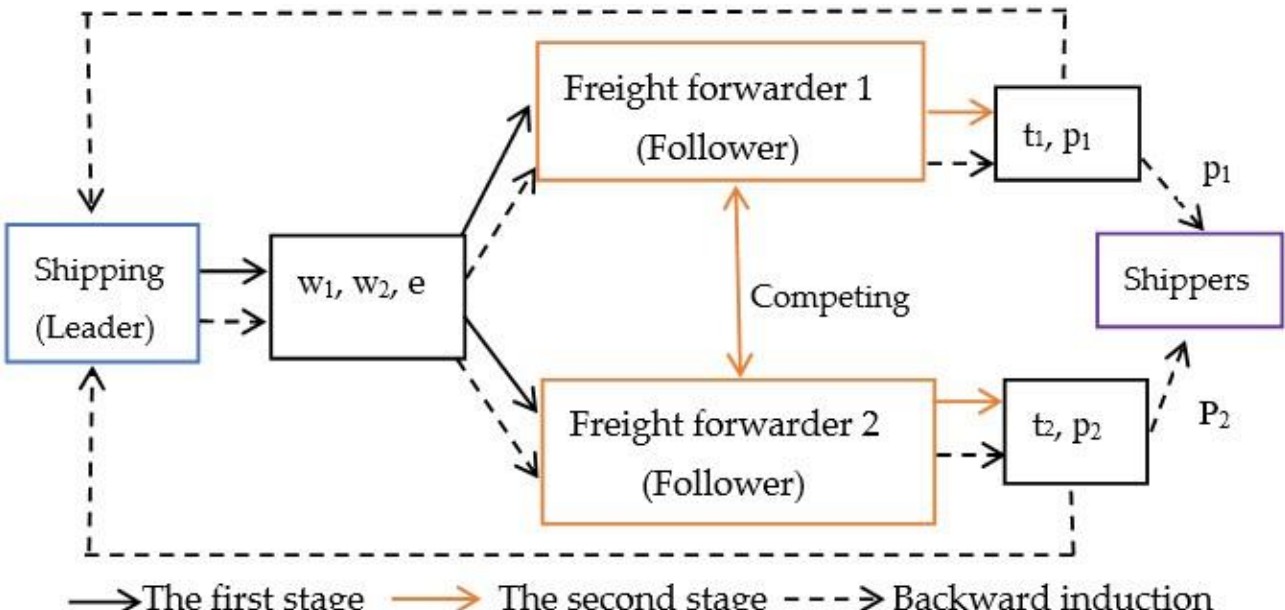

**Figure 1.** A diagram of the supply chain system structure.

Backward induction is applied to derive optimal results, which is extensively used to solve sequential game models.

Taking the first-order partial derivatives of Equation (7) with respect to $p_1$ and $t_1$, respectively, we can obtain the first-order conditions to maximize Equation (7):

$$\begin{cases} k - p_1 + \mu p_2 + \lambda e + \eta t_1 - (p_1 - w_1) = k - 2p_1 + \mu p_2 + \lambda e + \eta t_1 + w_1 = 0 \\ (p_1 - w_1)\eta - 2\beta t_1 = 0 \end{cases} \tag{10}$$

Taking the second-order derivatives, we can obtain the Hessian matrix:

$$H_0 = \begin{bmatrix} -2 & \eta \\ \eta & -2\beta \end{bmatrix} \tag{11}$$

Obviously, the second-order derivative satisfies that $\frac{\partial^2 \pi_{f_1}}{\partial p_1^2} = -2 < 0$, and based on Assumption 1 we have $\det(H_0) = 4\beta - \eta^2 > 0$. Thus, solving Equation (11) yields the following two reactive equations for Forwarder 1:

$$p_1 = \frac{b_1 - \mu p_2}{-2 + B} \tag{12}$$

$$t_1 = B_0(p_1 - w_1) = B_0 \frac{-b_1 - \mu p_2}{-2 + B} - B_0 w_1 \tag{13}$$

in which

$$b_1 = -k - \lambda e - (1 - B)w_1 \tag{14}$$

Taking the first-order partial derivatives of Equation (8) with respect to $p_2$ and $t_2$, respectively, we can obtain first-order conditions to maximize Equation (8):

$$\begin{cases} k - p_2 + \mu p_1 + \lambda e + \eta t_2 - (p_1 - w_1) = k - 2p_2 + \mu p_1 + \lambda e + \eta t_2 + w_2 = 0 \\ (p_2 - w_2)\eta - 2\beta t_2 = 0 \end{cases} \tag{15}$$

Hence, the Hessian matrix is still given as $H_0$. Therefore, the profit function is jointly concave on $p_2$ and $t_2$. Thus, solving Equation (15) yields the following two reactive equations for Forwarder 2:

$$p_2 = \frac{b_2 - \mu p_1}{-2 + B} \tag{16}$$

$$t_2 = B_0(p_2 - w_2) = B_0 \frac{-b_2 - \mu p_1}{-2 + B} - B_0 w_2 \tag{17}$$

where

$$b_2 = -k - \lambda e - (1 - B)w_2 \tag{18}$$

The following Lemma 1 can be derived under Assumption 2, which summarizes the forwarders' best response to the shipping prices given by the shipping company.

For the sake of exposition, the optimal solutions to the model are represented by the superscript $*$.

**Lemma 1.** *The following results hold true under Assumption 2:*

*(1)   the unique optimal decision pair $(p_1^*(w_1, e),\ t_1^*(w_1, e))$ of Forwarder 1 in reaction to $w_1$ and $e$ is given by*

$$p_1^*(w_1, e) = \frac{(-2 + B)b_1 - \mu b_2}{(-2 + B)^2 - \mu^2} \tag{19}$$

$$t_1^*(w_1, e) = B_0 \cdot \frac{(-2 + B)[-k - \lambda e + w_1] - \mu b_2 + \mu^2 w_1}{(-2 + B)^2 - \mu^2} \tag{20}$$

(2)   the unique optimal decision pair $(p_2^*(w_2, e),\ t_2^*(w_2, e))$ of Forwarder 2 in reaction to $w_2$ and $e$ is given by

$$p_2^*(w_2, e) = \frac{(-2 + B)b_2 - \mu b_1}{(-2 + B)^2 - \mu^2} \tag{21}$$

$$t_2^*(w_2, e) = B_0 \cdot \frac{(-2 + B)[-k - \lambda e + w_2] - \mu b_1 + \mu^2 w_2}{(-2 + B)^2 - \mu^2} \tag{22}$$

To simplify the discussion in the rest of the paper, we denote

$$d = (2 - B + \mu)k + c[-2 + B + \mu^2 + \mu(1 - B)] = k(2 - B + \mu) + c[-2 + B(1 - \mu) + \mu + \mu^2] \tag{23}$$

and denote

$$C = -2 + B + \mu^2,\ D = \mu(1 - B) \tag{24}$$

Then, we have the following Lemmas.

**Lemma 2.** *Under Assumption 3, we have*

$$d = (2 - B + \mu)[k - c(1 - \mu)] > 0 \tag{25}$$

**Lemma 3.** *Under Assumption 3, the following properties hold:*

$$C + D = -(2 - B + \mu)(1 - \mu) < 0 \tag{26}$$

$$C - D = -(2 - B - \mu)(1 + \mu) < 0 \tag{27}$$

Based on the previous preliminaries, we have the following proposition detailing the equilibrium results for the two-stage game.

**Proposition 1.** *When the shipping company and two forwarders act in a completely self-interested manner and if the parameters of the supply chain satisfy*

$$\lambda^2 < \frac{-2\alpha(C + D)(2 - B - \mu)}{2 - B + \mu} \tag{28}$$

*then*

(i)   *the unique optimal shipping prices and brand value of the shipping company are given by:*

$$w_1^* = w_2^* = c + \frac{-d\alpha(2 - B - \mu)}{\lambda^2(2 - B + \mu) + 2\alpha(C + D)(2 - B - \mu)} \tag{29}$$

$$e^* = \frac{-d\lambda}{\lambda^2(2 - B + \mu) + 2\alpha(C + D)(2 - B - \mu)} \tag{30}$$

*and* $w_i^* > c (i = 1, 2)$, $e^* > 0$;

(ii)   *the unique optimal freight service prices of the freight forwarders are as follows:*

$$p_1^* = p_2^* = c + \frac{\alpha[k - c(1 - \mu)] \cdot (C + D) - d\alpha(2 - B - \mu)}{\lambda^2(2 - B + \mu) + 2\alpha(C + D)(2 - B - \mu)} \tag{31}$$

*and* $p_i^* > w_i^*$, $i = 1, 2$;

*(iii)*    *the unique optimal effort levels of the extending brand value are expressed as*:

$$t_1^* = t_2^* = \frac{B_0 \cdot \alpha(C+D)[k-c(1-\mu)]}{\lambda^2(2-B+\mu) + 2\alpha(C+D)(2-B-\mu)} \tag{32}$$

*and* $t_i^* > 0$, $i = 1, 2$;

*(iv)*    *the market demands under the optimal decisions are*:

$$Q_1^* = Q_2^* = \frac{\alpha(C+D)[k-c(1-\mu)]}{\lambda^2(2-B+\mu) + 2\alpha(C+D)(2-B-\mu)} \tag{33}$$

*and* $Q_1^*, Q_2^* > 0$;

*(v)*    *the maximal profit of the shipping company is*:

$$\pi_s^* = \frac{-\alpha d[k-c(1-\mu)]}{\lambda^2(2-B+\mu) + 2\alpha(C+D)(2-B-\mu)} \tag{34}$$

*and* $\pi_s^* > 0$;

*(vi)*    *the maximal profits of the freight forwarders are*:

$$\begin{aligned}
\pi_{f_1}^* = \pi_{f_2}^* &= \frac{\alpha^2(1-\beta B_0^2)[(2-B+\mu)(\mu-1)]^2[k-c(1-\mu)]^2}{[\lambda^2(2-B+\mu)-2\alpha(2-B+\mu)(1-\mu)(2-B-\mu)]^2} \\
&= \frac{\alpha^2(1-\beta B_0^2)(1-\mu)^2[k-c(1-\mu)]^2}{[\lambda^2-2\alpha(1-\mu)(2-B-\mu)]^2}
\end{aligned} \tag{35}$$

*and* $\pi_{f_1}^*$, $\pi_{f_2}^* > 0$.

Condition (28) indicates that the brand preference of shippers does not exceed the given range.

The equilibrium solutions indicate that even though there is competition between freight Forwarders 1 and 2, the shipping company, being the market leader, will set the same shipping prices for freight Forwarders 1 and 2 to promote coordination and fairness within the SCSCS, assuming all members are all completely self-interested. Since both freight forwarders accept the same shipping prices, they exert equal effort to extend the brand value and charge the shippers the same freight service prices, thereby ensuring internal stability and coordination of the SCSCS.

It is noteworthy that the optimal shipping prices presented in Equation (29) can be divided into two components: cost and non-cost. Similarly, Equation (31) shows that the optimal freight service prices also consist of two parts: shipping price and non-shipping price. Not surprisingly, we have $w_i^* > c$ and $p_i^* > w_i^*$. Because $w_i^* - c$ and $p_i^* - w_i^*$ represent the marginal profits of the shipping company and freight forwarders, respectively, $w_i^* > c$ and $p_i^* > w_i^*$ are the premises of the normal operations for the supply chain.

According to Formulas (29) and (31), we have $\frac{dw_i^*}{d\lambda} > 0$ and $\frac{dp_i^*}{d\lambda} > 0$. Thus, the optimal shipping prices of the shipping company and optimal freight service prices of the freight forwarders are positively correlated with the shippers' brand preference. This indicates that high brand preference of shippers brings in high shipping prices and high freight service prices.

Observed from Formula (30), the equilibrium solution of the shipping company' brand value has a positive association with the shippers' brand preference according to $\frac{de^*}{d\lambda} > 0$. This observation implies that enhancing the brand awareness of the shippers helps to increase the brand value of the shipping company.

From Formulas (32) and (3), the optimal effort levels of the extending brand value for freight forwarders are positively correlated with $\eta$ and negatively correlated with $\beta$. This indicates that the higher the sensitivity of the market demand to the efforts of extending brand value, the more willing freight forwarders are to make such efforts, while the higher

the effort cost incurred by freight forwarders in extending brand value, the less willing they are to make such efforts.

We immediately find from (29), (30), and (31) that $w_1^* = w_2^* = c + \frac{\alpha(2-B-\mu)}{\lambda}e^*$ and $p_1^* = p_2^* = \frac{\alpha[k-c(1-\mu)]\cdot(C+D)}{-d\lambda}e^* + w_i^*$. This means that the optimal shipping prices of the shipping company and optimal freight service prices of the freight forwarders are positively correlated with the brand value of the shipping company. Meanwhile, the above two formulas reflect the pricing bases of the shipping company and the freight forwarders, respectively. This suggests that high brand value brings in high shipping prices and freight service prices, and then inspires that the shipping company should improve its brand value to win the consumers with brand preference.

According to Formulas (32), Formula (33) can be rewritten as $Q_1^* = Q_2^* = t_i^*/B_0$. This shows that market demands are positively related to the freight forwarders' effort levels. That is, the higher the freight forwarders' effort levels, the more effective they are in stimulating market demands. Because the increases of their effort levels can attract more customers, the market demands will increase. From a management perspective, this implies that the increase in the investment to extend the brand value can increase market demands, which is beneficial for the increase of profits for all parties.

Formula (33) can be rewritten as $Q_1^* = Q_2^* = \frac{\alpha(C+D)[k-c(1-\mu)]}{-d\lambda}e^*$. This shows that the increase in the shipping company' brand value can indeed increase market demand. Formula (34) can be rewritten as $\pi_s^* = \frac{-\alpha d[k-c(1-\mu)]}{-d\lambda}e^*$. This indicates that the rise in profit for the shipping company is indeed obtained by increasing its brand value. These provide the managerial insight that the shipping company should increase its brand investment.

According to Formulas (33) and (34), it is easy to see that market demands and the shipping company' profit are positively correlated with the brand preference of shippers. These further demonstrate the importance of brand investment for the shipping company.

The above findings suggest that the shipping company' brand value and the shippers' brand preference have positive impacts on equilibrium results. Consequently, all the participants are willing to invest in brand building or extending the brand value.

From Formula (35), we first find that each freight forwarder in the SCSCS earns equal profit, and this might be due to the fact that all freight forwarders are equally important to the supply chain, even though there is competition between the freight forwarders. Second, the freight forwarders' profits have a positive association with the brand preference of shippers. That is, the brand preference of shippers also benefits the freight forwarders. This inspires freight forwarders to improve the effort levels extending brand value.

Furthermore, we will show that these findings remain valid under the scenario with altruism preference.

### 3.2. Decisions under the Altruistic Preference of Each Forwarder

In view of the increasingly complex economic environment and fierce market competition background, the freight forwarders would be concerned about the profits of the shipping company in decision making to promote cooperation with the shipping company with a good brand reputation while considering their own profits. In this case, the shipping company aims to maximize their own profit, and the freight forwarders with altruistic preferences take the maximization of their own utilities as decision-making goals, rather than maximization of their own profit to determine the service prices and effort levels. According to the description of altruistic preference in references [9,12–15], when the two freight forwarders possess altruistic preferences, the utility functions of the shipping company and Forwarders 1 and 2 can be respectively expressed as follows:

$$U_s(w_1, w_2, e) = \pi_s = (w_1 - c)(k - p_1 + \mu p_2 + \lambda e + \eta t_1) \\ + (w_2 - c)(k - p_2 + \mu p_1 + \lambda e + \eta t_2) - \alpha e^2 \tag{36}$$

$$
\begin{aligned}
U_{f_1} &= \pi_{f_1} + \varepsilon \pi_s \\
&= (p_1 - w_1 + \varepsilon(w_1 - c))(k - p_1 + \mu p_2 + \lambda e + \eta t_1) \\
&\quad + \varepsilon(w_2 - c)(k - p_2 + \mu p_1 + \lambda e + \eta t_2) - \varepsilon \alpha e^2 - \beta t_1^2
\end{aligned}
\tag{37}
$$

$$
\begin{aligned}
U_{f_2} &= \pi_{f_2} + \varepsilon \pi_s \\
&= (p_2 - w_2 + \varepsilon(w_2 - c))(k - p_2 + \mu p_1 + \lambda e + \eta t_2) \\
&\quad + \varepsilon(w_1 - c)(k - p_1 + \mu p_2 + \lambda e + \eta t_1) - \alpha \varepsilon e^2 - \beta t_2^2
\end{aligned}
\tag{38}
$$

The utility function of each freight forwarder includes not only its own profit, but also part of the profit of the shipping company, that is, each freight forwarder has altruistic preference behavior because the interest of the partner is considered. The degree of the altruistic preference is denoted as $\varepsilon$ ($0 \le \varepsilon \le 1$), where a higher value of $\varepsilon$ indicates a greater tendency for the freight forwarders to act altruistically, which means they will make more effort to increase the shipping company's profits. When $\varepsilon$ is 0, the freight forwarders have no altruistic preference towards the shipping company and are entirely self-interested. When $\varepsilon$ is 1, the freight forwarders view the increase in profits for themselves and the shipping company as equally important, indicating that they are completely altruistic decision makers.

The decision sequence is the same as that in the previous subsection. All supply chain members make decisions to maximize their utility function.

First, by calculating the first-order partial derivatives of Equation (37) with respect to $p_1$ and $t_1$, we can obtain the first-order conditions to maximize Equation (37):

$$
\begin{cases}
k - 2p_1 + \mu p_2 + \lambda e + \eta t_1 + \overline{w}_1 + \varepsilon \mu (w_2 - c) = 0 \\
(p_1 - \overline{w}_1)\eta - 2\beta t_1 = 0
\end{cases}
\tag{39}
$$

where

$$
\overline{w}_1 = w_1 - \varepsilon(w_1 - c) = w_1 - c - \varepsilon(w_1 - c) + c = (w_1 - c)(1 - \varepsilon) + c
\tag{40}
$$

Then, taking the second-order derivatives, we obtain the Hessian matrix $H_0$ of the utility function $U_{f_1}$, which is identical to the one presented Equation (11). This confirms that $U_{f_1}$ is jointly concave with respect to $p_1$ and $t_1$. Therefore, by solving Equation (39) yields the following two reactive equations for Forwarder 1:

$$
p_1 = \frac{b_3 - \mu p_2}{-2 + B}
\tag{41}
$$

$$
t_1 = B_0(p_1 - \overline{w}_1) = B_0 \frac{-b_3 - \mu p_2}{-2 + B} - B_0 \overline{w}_1
\tag{42}
$$

in which $B_0$ is defined in Equation (12), and

$$
b_3 = -k - \lambda e + (B - 1)\overline{w}_1 - \varepsilon \mu (w_2 - c)
\tag{43}
$$

Taking the first-order partial derivatives of Equation (38) with respect to $p_1$ and $t_1$, respectively, we can obtain the first-order conditions to maximize Equation (38):

$$
\begin{cases}
k - 2p_2 + \mu p_1 + \lambda e + \eta t_2 + \overline{w}_2 + \varepsilon \mu (w_1 - c) = 0 \\
(p_2 - \overline{w}_2)\eta - 2\beta t_2 = 0
\end{cases}
\tag{44}
$$

where

$$
\overline{w}_2 = w_2 - \varepsilon(w_2 - c) = w_2 - c - \varepsilon(w_2 - c) + c = (w_2 - c)(1 - \varepsilon) + c.
\tag{45}
$$

Hence, the Hessian matrix of $U_{f_2}$ is still $H_0$ given by Equation (11). Therefore, the utility function is jointly concave on $p_2$ and $t_2$. Thus, solving Equation (44) yields the following two reactive equations for Forwarder 2:

$$p_2 = \frac{b_4 - \mu p_1}{-2 + B} \tag{46}$$

$$t_2 = B_0(p_2 - \overline{w}_2) = B_0 \frac{-b_4 - \mu p_1}{-2 + B} - B_0 \overline{w}_2 \tag{47}$$

where

$$b_4 = -k - \lambda e + (B - 1)\overline{w}_2 - \varepsilon\mu(w_1 - c) \tag{48}$$

Based on the above discussions, the optimal decisions made by both forwarders in response to the shipping prices $(w_1, w_2)$ and the brand value $(e)$ set by the shipping company are characterized in the following context.

For the sake of exposition, the optimal solutions to the model with altruistic preference are denoted by the superscript $**$.

**Lemma 4.** *When each forwarder has an altruistic preference, the following results can be obtained if Assumption 2 is met.*

(1)　*the unique optimal decision pair $(p_1^{**}(w_1, e),\ t_1^{**}(w_1, e))$ of Forwarder 1 in reaction to $w_1$ and $e$ is given by*

$$p_1^{**}(w_1, e) = \frac{(-2 + B)b_3 - \mu b_4}{(-2 + B)^2 - \mu^2} \tag{49}$$

$$t_1^{**}(w_1, e) = B_0 \cdot \frac{(-2 + B)[-k - \lambda e + \overline{w}_1 - \varepsilon\mu(w_2 - c)] - \mu b_4 + \mu^2 \overline{w}_1}{(-2 + B)^2 - \mu^2} \tag{50}$$

(2)　*the unique optimal decision pair $(p_2^{**}(w_2, e),\ t_2^{**}(w_2, e))$ of Forwarder 2 in reaction to $w_2$ and $e$ is given by*

$$p_2^{**}(w_2, e) = \frac{(-2 + B)b_4 - \mu b_3}{(-2 + B)^2 - \mu^2} \tag{51}$$

$$t_2^{**}(w_2, e) = B_0 \cdot \frac{(-2 + B)[-k - \lambda e + \overline{w}_2 - \varepsilon\mu(w_1 - c)] - \mu b_3 + \mu^2 \overline{w}_2}{(-2 + B)^2 - \mu^2} \tag{52}$$

To derive the subsequent main results, some lemmas should be introduced first. Let

$$\begin{aligned} M &= (-2 + B)(1 - \varepsilon) + \mu^2 \\ N &= \mu(B - 1)(3\varepsilon - \varepsilon B - 1) + \mu^3 \varepsilon \end{aligned} \tag{53}$$

Thus, we have the following arguments.

**Lemma 5.** *Under Assumption 2, the following properties hold:*

(1)　$M + N = s_1 \varepsilon + s_2$, *where*

$$\begin{aligned} s_1 &= 2 - B + \mu(B - 1)(3 - B) + \mu^3 = (2 - B + \mu)[(1 - \mu)^2 + \mu B] > 0 \\ s_2 &= -2 + B + \mu^2 - \mu(B - 1) = C + D = -(2 - B + \mu)(1 - \mu) < 0 \end{aligned} \tag{54}$$

(2)　$M + N < 0$, *when* $\varepsilon < -s_2/s_1$.

(3)　$M - N = s_3 \varepsilon + s_4$, *where*

$$\begin{aligned} s_3 &= (2 - B - \mu)[(1 + \mu)^2 - \mu B] > 0 \\ s_4 &= -(2 - B - \mu)(1 + \mu) < 0 \end{aligned} \tag{55}$$

(4)　$M - N < 0$, *when* $\varepsilon < -s_4/s_3$.

*Define*

$$s_5 = 2s_1 - (2 - B + \mu) + (2 - B + \mu)(2 - B - \mu) \tag{56}$$

**Lemma 6.** *Under Assumption 2, we have $s_5 > 0$.*

Based on the aforementioned preliminaries, we present the following proposition that elaborates the equilibrium results for the two-stage game.

**Proposition 2.** *When two forwarders have altruistic preferences and if the parameters of the supply chain satisfy the following two conditions:*

$$\varepsilon < \varepsilon_0 = \min\left\{ \frac{-s_2}{s_1}, \frac{-s_4}{s_3}, \frac{2 - B - \mu^2}{2 - B}, \frac{-s_2}{s_5} \right\} \tag{57}$$

$$\lambda^2 < \frac{-2\alpha(M + N)(2 - B - \mu)}{2 - B + \mu} \tag{58}$$

*then*

*(i)* *the unique optimal shipping prices and brand value of the shipping company are*

$$w_1^{**} = w_2^{**} = c + \frac{-d\alpha(2 - B - \mu)}{\lambda^2(2 - B + \mu) + 2\alpha(M + N)(2 - B - \mu)} \tag{59}$$

*and $w_i^{**} > c$ ($i = 1, 2$), $e^{**} > 0$;*

$$e^{**} = \frac{-d\lambda}{\lambda^2(2 - B + \mu) + 2\alpha(M + N)(2 - B - \mu)} \tag{60}$$

*(ii)* *the unique optimal freight service prices are*

$$p_1^{**} = p_2^{**} = c + \frac{\alpha[k - (1 - \mu)c] \cdot [2s_1\varepsilon + s_2 - (2 - B + \mu)\varepsilon] - d\alpha(2 - B - \mu)(1 - \varepsilon)}{\lambda^2(2 - B + \mu) + 2\alpha(M + N)(2 - B - \mu)} \tag{61}$$

*and $p_i^{**} > w_i^{**}$, $i = 1, 2$;*
*(iii)* *the unique optimal effort levels for extending the brand value are*

$$t_1^{**} = t_2^{**} = \frac{\alpha B_0 \cdot [k - (1 - \mu)c] \cdot [2s_1\varepsilon + s_2 - \varepsilon(2 - B + \mu)]}{\lambda^2(2 - B + \mu) + 2\alpha(M + N)(2 - B - \mu)} \tag{62}$$

*and $t_i^{**} > 0$, $i = 1, 2$;*
*(iv)* *the market demands under the optimal decisions are*

$$Q_1^{**} = Q_2^{**} = \frac{\alpha(M + N)[k - c(1 - \mu)]}{\lambda^2(2 - B + \mu) + 2\alpha(M + N)(2 - B - \mu)} \tag{63}$$

*and $Q_1^{**}, Q_2^{**} > 0$;*
*(v)* *the maximal profit of the shipping company is*

$$\pi_s^{**} = \frac{-\alpha d[k - c(1 - \mu)]}{\lambda^2(2 - B + \mu) + 2\alpha(M + N)(2 - B - \mu)} \tag{64}$$

*and $\pi_s^{**} > 0$;*
*(vi)* *the maximal profits of the freight forwarders are*

$$\pi_{f_1}^{**} = \pi_{f_2}^{**} = \frac{\alpha^2 \cdot [k - (1 - \mu)c]^2 \cdot L}{[\lambda^2(2 - B + \mu) + 2\alpha(M + N)(2 - B - \mu)]^2} \tag{65}$$

*where*

$$L = (1 - \beta \cdot B_0{}^2)(M+N)^2 + \varepsilon E(M+N) - \beta \cdot B_0{}^2[s_1\varepsilon - \varepsilon(2 - B + \mu)]^2$$
$$E = (1 - B)[s_1 - (2 - B + \mu)] + (2 - B + \mu)(2 - B - \mu) = (1 - B)s_1 - s_2 \tag{66}$$

Conditions (57) and (58) imply that the altruistic preference coefficient of the freight forwarders and the brand preference of the shippers are kept in a specific range, respectively.

For the formulas in Proposition 2, their meaning and management implications are similar to those in Proposition 2, and the impacts of the brand preference and the altruistic preference on equilibrium solutions is explained in the next section.

**Remark 1.** *It is worth mentioning that the proof of Proposition 2 is rather technical, and from the proof, we can see that the second leading principal minor can also be guaranteed to be positive when $M + N > 0$ and $M - N > 0$. However, this will lead to a contradiction, and the proof can be found in Supplementary Material S3. Thus, $M + N < 0$ and $M - N < 0$ are the necessary conditions for the existence of the equilibrium strategies.*

**Remark 2.** *In Proposition 2, we have pointed out that $\pi_s^{**} > 0$. Nonetheless, we find that $\pi_{f_1}^{**}(= \pi_{f_2}^{**}) > 0$ may not hold under the conditions of Proposition 2. Because we focus on the positive profits of all members in the supply chain at equilibrium, we thus establish tighter bounds on $\varepsilon$ for $\pi_{f_1}^{**}(= \pi_{f_2}^{**}) > 0$, as follows. This is, $\varepsilon$ needs to satisfy this constraint,*
*$\varepsilon < \varepsilon_1 = \min\left\{\frac{-s_4}{s_3}, \frac{2-B-\mu^2}{2-B}, \varepsilon_2\right\}$, where*

$$\varepsilon_2 = \frac{-s_2(2 - B)}{(2 - B)s_1 + E + \sqrt{E^2 + (2 - B)B[s_1 - (2 - B + \mu)]^2}} \tag{67}$$

*Here, E is given by Equation (66).*

We first derive Corollary 1 to show that $\pi_{f_1}$ and $\pi_{f_2}$ are positive when $\varepsilon < \varepsilon_1$, and then we explain that the condition for $\varepsilon$ in Corollary 1 is tighter than that in Proposition 2.

**Corollary 1.** *Assuming that $\varepsilon < \varepsilon_1$, and other conditions are the same as those in Proposition 2; consequently, we get $\pi_{f_1} = \pi_{f_2} > 0$.*

We can prove that the condition $\varepsilon < \varepsilon_1$ in Corollary 1 is tighter than $\varepsilon < \varepsilon_0$ in Proposition 2, and the proof can be found in Supplementary Material S4.

It can be inferred from Proposition 2 and Corollary 1 that if the freight forwarders are moderately altruistic, it is beneficial to form a cooperative alliance with the self-interested shipping company.

## 4. Analyses of the Equilibrium Results

We proceed to investigate how the optimal/equilibrium shipping prices, the brand value, the freight service prices, the brand extension efforts, the market demands, and the supply chain members' profits are affected by the model parameters in a sequential manner. Specifically, our analysis focuses on the impact of two key parameters, namely the degree of altruistic preferences among freight forwarders and the brand preference of shippers.

**Proposition 3.** *Assuming that the conditions in Proposition 2 hold, then*

*(i)   $w_i^{**}, e^{**}, p_i^{**}, t_i^{**}$ and $Q_i^{**}$ $(i = 1, 2)$ increase with $\lambda$ and $\varepsilon$.*
*(ii)  $w_i^{**} > w_i^*, e^{**} > e^*, p_i^{**} > p_i^*, t_i^{**} > t_i^*$ and $Q_i^{**} > Q_i^*$ $(i = 1, 2)$.*

Proposition 3 (i) implies that (1) shippers with a strong brand preferences lead to an increase in the shipping prices and freight service prices; (2) the stronger the shippers' brand willingness, the larger the market demand that can be generated with the forwarders'

efforts of extending the brand value, increasing their motivation towards such efforts. Consequently, the expanded market demand provides the foundation for a concurrent rise in the profit of the shipping company.

Proposition 3 (i) also implies that when the freight forwarders have stronger altruistic preferences, they are willing to prioritize the shipping company's profits in maintaining a mutually beneficial business relationship. However, they also need to raise the prices of their freight services to offset the cost of their altruistic behaviors.

Proposition 3 (ii) reveals the following management implications.

The freight forwarders' altruistic preferences increase their market demand, which further motivates the shipping company to invest in brand building. As the altruistic preferences of the freight forwarders strengthen, they exert more effort in extending the shipping company's brand value. Proposition 3 also suggests that when the freight forwarders are attentive to the shipping company's profits, they can accept relatively higher shipping prices. However, this altruistic preference leads to narrower profit margins for the freight forwarders themselves. As a result, they need to make additional efforts to extend the brand value of the shipping company to stimulate shipping demand and recoup their costs. Consequently, freight forwarders are inclined to raise the price of their freight services offered to shippers. The shipping company utilizes a portion of the profits transferred by the freight forwarders to enhance its brand value. Through the collaborative efforts of the shipping company and freight forwarders, the expanded market demand serves as the foundation for a simultaneous increase in their profits.

**Proposition 4.** *Assuming that the conditions in Proposition 2 hold, then $\pi_s^{**}$ increase with $\lambda$ and $\varepsilon$, and $\pi_s^{**} > \pi_s^*$.*

We next examine the effects of the sensitivity degree of the market demand to the brand value and the degree of the altruistic preference of the freight forwarders on the freight forwarders' equilibrium profits. For this, let

$$F = \lambda^2(2 - B + \mu) + 2\alpha(M + N)(2 - B - \mu) \tag{68}$$

**Proposition 5.** *Assuming that conditions in Collary 1 are met, then we have*

(1) $\pi_{f_1}^{**}$ and $\pi_{f_2}^{**}$ *increase with $\lambda$.*

(2) $\pi_{f_1}^{**}$ and $\pi_{f_2}^{**}$ *increase with $\varepsilon$, $\pi_{f_1}^{**} > \pi_{f_1}^*$ and $\pi_{f_2}^{**} > \pi_{f_2}^*$ provided that one of the following conditions holds: (a) $L_d > 0$; (b) $L_d < 0$ and $L_d \cdot F < 4L \cdot \alpha s_1(2 - B - \mu)$.*

(3) $\pi_{f_1}^{**}$ and $\pi_{f_2}^{**}$ *decrease with $\varepsilon$, $\pi_{f_1}^{**} < \pi_{f_1}^*$, and $\pi_{f_2}^{**} < \pi_{f_2}^*$, provided that the following conditions hold: (c) $L_d < 0$; (d) $L_d \cdot F > 4L \cdot \alpha s_1(2 - B - \mu)$. where $L$ is defined by Equations (63); $L_d = 2l_1\varepsilon + l_2$ with $l_1$ and $l_2$ defined as $l_1 = s_1{}^2(2 - 2B) - s_1 s_2 + s_1 B(2 - B + \mu) - \beta \cdot B_0{}^2(2 - B + \mu)^2$, and $l_2 = (3 - 2B)s_1 s_2 - s_2^2$, respectively.*

Propositions 4 and 5 suggest that a shipping company's strong brand investment willingness can have a positive impact on all decision-making parties involved in the supply chain, ultimately leading to increased profits for the entire SCSCS. When the freight forwarders have altruistic preferences, their profits increase if either conditions (a) or (b) are met, motivating them to further extend the brand value. Therefore, the freight forwarders' altruistic preferences can increase the profits of other SCSCS members and the entire SCSCS. However, if conditions (c) and (d) are met, freight forwarders' profits will decrease.

Finally, the theoretical results in Propositions 3 to 5 are summarized in Table 3.

**Table 3.** Results in Propositions 3 to 5.

|  | $w_i^{**} (> w_i^*)$ | $e^{**} (> e^*)$ | $p_i^{**} (> p_i^*)$ | $t_i^{**} (> t_i^*)$ | $Q_i^{**} (> Q_i^*)$ | $\pi_s^{**} (> \pi_s^*)$ | $\pi_{f_i}^{**}$ |
|---|---|---|---|---|---|---|---|
| $\lambda$ | + | + | + | + | + | + | + |
| $\varepsilon$ | + | + | + | + | + | + | / |

Note: "+" indicates positive correlation, "/" indicates indefinite, and the specific conditions for increase or decrease are explained in Proposition 5.

## 5. Simulation and Numerical Analysis

In this section, to verify the conclusions above and illustrate the impacts of the parameters on equilibrium results, a numerical example is demonstrated.

Herein, we only provide methods for determining the basic parameter values. The detailed data collection and calculation basis can be found in Supplementary Material S5. Based on the container shipping quotation (USD 250 per standard container) from a specific port of departure to a specific port of destination, we can estimate that the shipping company's marginal cost is approximately USD 150 according to the profit rates. The basic market demand is calculated based on the total number of freight forwarders and container throughput in the city located at the departure port, estimating that there are $k = 135$ standard container per year. So, $k$ satisfies Assumption 3. The cost coefficients are selected as $\alpha = 4$ [4] and $\beta = 2.5$ [44]. A lower competition coefficient, $\mu = 0.3$, is chosen based on the literature [40]. Using Assumption 2, take $\eta = 2.2$. That is, $\eta < \sqrt{2\beta(2 - \mu)}$. In conclusion, the basic parameters are set as $c = 150$, $k = 135$, $\alpha = 4$, $\beta = 2.5$, $\eta = 2.2$, and $\mu = 0.3$. From Remark 2, we have $\varepsilon \in (0, 0.328).\lambda \in J = (0, 1.61)$, as directly calculated using Formula (58).

Firstly, we analyze the effects of altruistic preference on equilibrium decisions. To do so, $\varepsilon$ is set as an independent variable and $\lambda = 1.2 \in J$. By calculation, we know that the conditions (c) and (d) hold in Proposition 5. Figure 2 illustrates that the shipping company's shipping prices and the freight forwarders' freight service prices both increase with $\varepsilon$. This is due to the fact that when the freight forwarders give a high importance to the profits of the shipping company, it means that they are willing to transfer a greater portion of their profits to the shipping company. Therefore, they are more likely to accept the relatively high shipping prices offered by the shipping company. However, the freight forwarders also aim to maintain their profits; hence they increase the freight service prices offered to shippers.

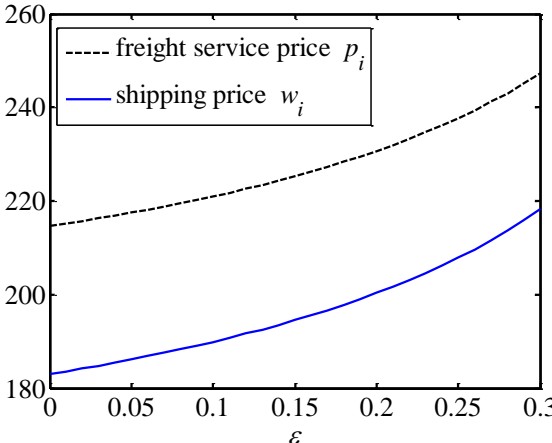

**Figure 2.** Effects of altruism preference on the shipping price and the freight service price.

Figure 3 shows that the brand value of shipping companies is positively related to altruism preference. This is because the shipping company can use the revenue transferred to it by the freight forwarder to make more brand investments and improve its own brand value. From Figures 4 and 5, the freight forwarders' effort levels and market demands are both positively correlated with $\varepsilon$. This is because they have to make more efforts of

extending the brand value to stimulate shipping demands and compensate their costs due to the freight forwarders' altruistic preference. As depicted in Figure 6, with the increase in $\varepsilon$, the profit of the shipping company and the overall profit of the supply chain increase, while the profits of the freight forwarders decline. These findings indicate that the freight forwarders' altruistic preferences positively impact the supply chain alliance, and in turn the alliance can compensate them for their altruistic behavior through profit redistribution.

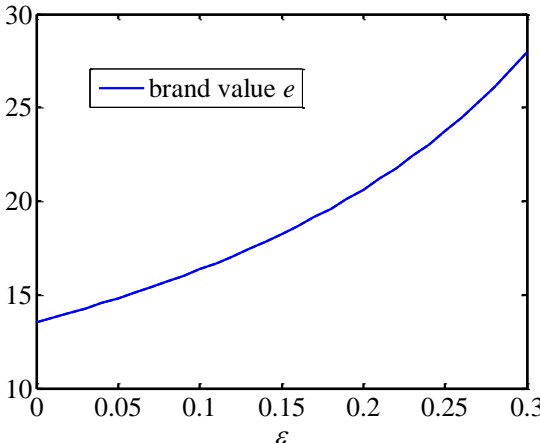

**Figure 3.** Effect of altruism preference on the brand value.

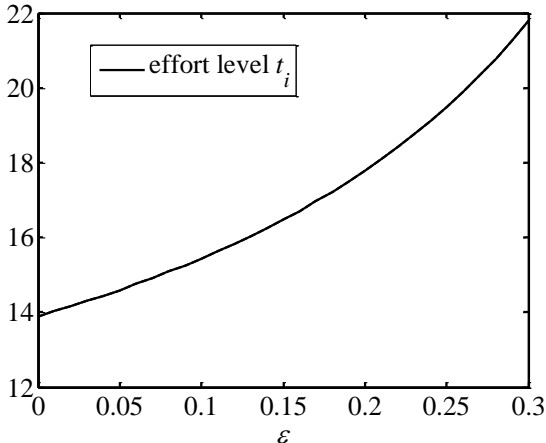

**Figure 4.** Effect of altruism preference on the effort level.

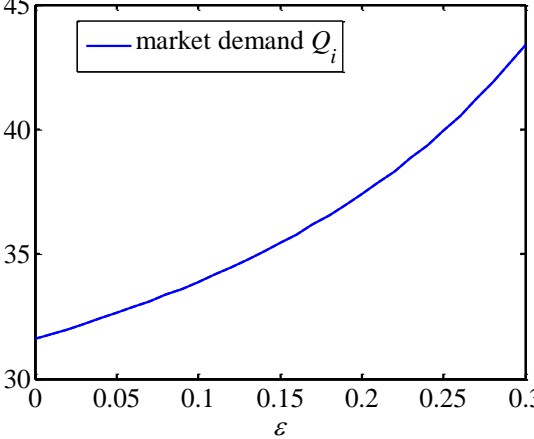

**Figure 5.** Effect of altruism preference on the market demand.

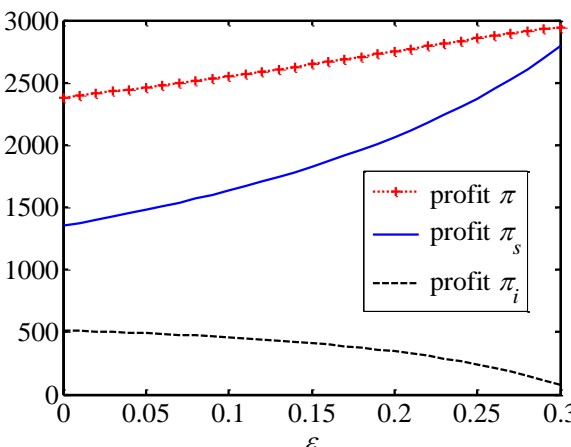

**Figure 6.** Effects of altruism preference on the profits.

Secondly, we examine the joint effects of the shippers' brand preference and altruism preference on equilibrium decisions. Here we only draw the results of $\lambda$ varying from 0 to 1.5. Thus, we can perform a sensitivity analysis on $\lambda$ and $\varepsilon$ and present the results in Figures 7–11. Figures 7 and 8 reveal that the shipping price and brand value of the shipping company and the freight service price offered by the freight forwarders are raised $\varepsilon$ under the joint effects of $\lambda$ and $\varepsilon$.This indicates that the shipping company invests more in brand building when there are stronger shippers' brand preferences or freight forwarders' altruism preferences. In this case, the shipping company raises the shipping prices offered to freight forwarders. Facing an increasing cost, freight forwarders naturally charge shippers higher freight service prices. In addition, the freight forwarders' effort levels and market demands are also positively correlated with $\lambda$ and $\varepsilon$, as shown in Figures 9 and 10. These findings suggest that stronger shippers' brand preference and freight forwarders' altruism preferences lead to larger market demands obtained by the freight forwarders' same effort level of extending the brand value, which will also improve the enthusiasm for the freight forwarders to extend the brand value. From Figure 11, the profit of the shipping company and the overall profit of the supply chain increase with the increase of $\lambda$. Overall, the results indicate that the shippers' brand preference can promote all decision-makers' willingness to participate in brand building, thereby leading to a higher overall profit of the supply chain.

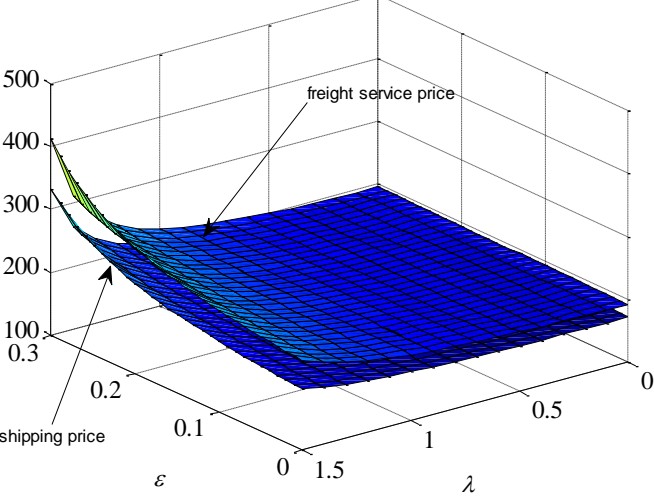

**Figure 7.** Effects of brand preference and altruism preference on the shipping price and the freight service price.

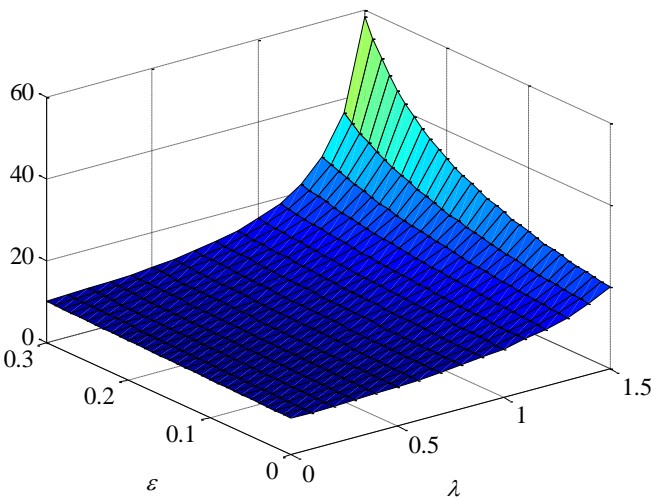

**Figure 8.** Effect of brand preference and altruism preference on the brand value.

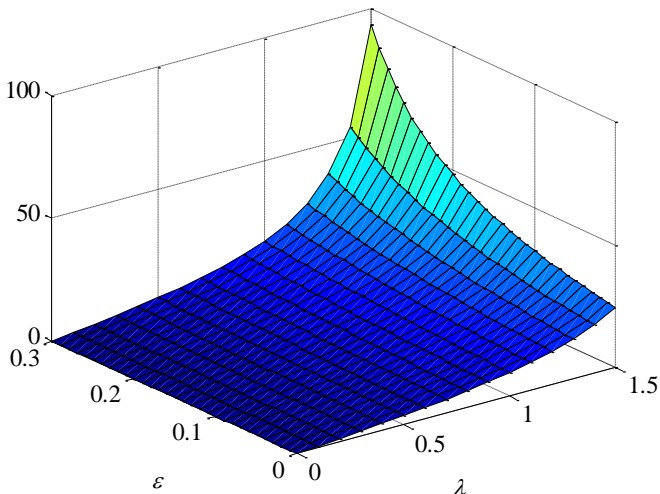

**Figure 9.** Effect of brand preference and altruism preference on the effort level.

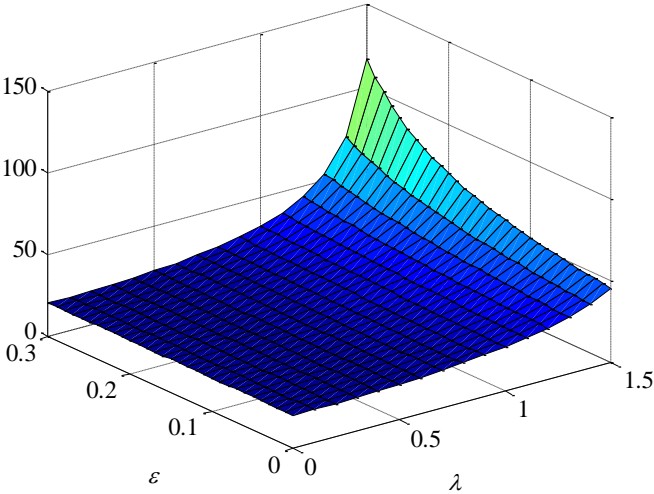

**Figure 10.** Effect of brand preference and altruism preference on the market demand.

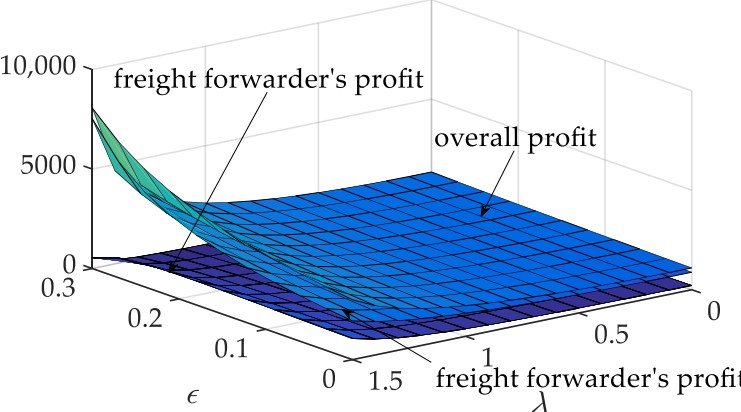

**Figure 11.** Effects of brand preference and altruism preference on the profits.

Thirdly, we analyze the effects of competition intensity on equilibrium decisions by varying the competition coefficient $\mu$ from 0 to 1 while maintaining constant $\varepsilon$ (we take that $\varepsilon = 0.2$). The results of the sensitivity analysis are presented in Figures 12–16. It is worth noting that when $\mu$ exceeds 0.5, the optimal prices and other factors become irregular. Therefore, we only present the results for $\mu$ less than 0.7.

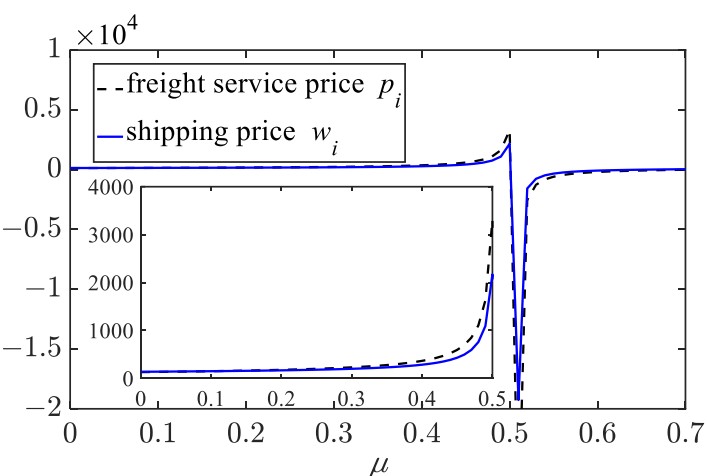

**Figure 12.** Effects of competition coefficient on the shipping price and the freight service price.

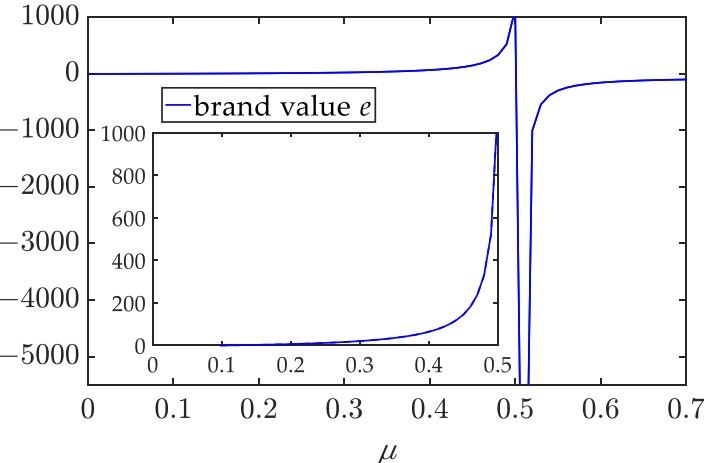

**Figure 13.** Effect of competition coefficient on the brand value.

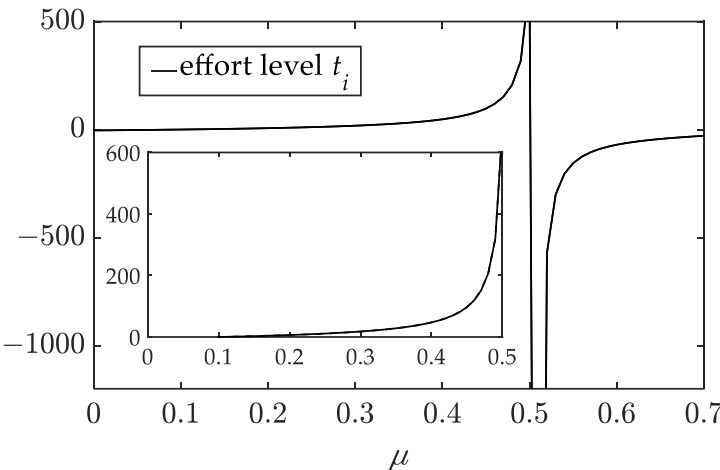

**Figure 14.** Effect of competition coefficient on the effort level.

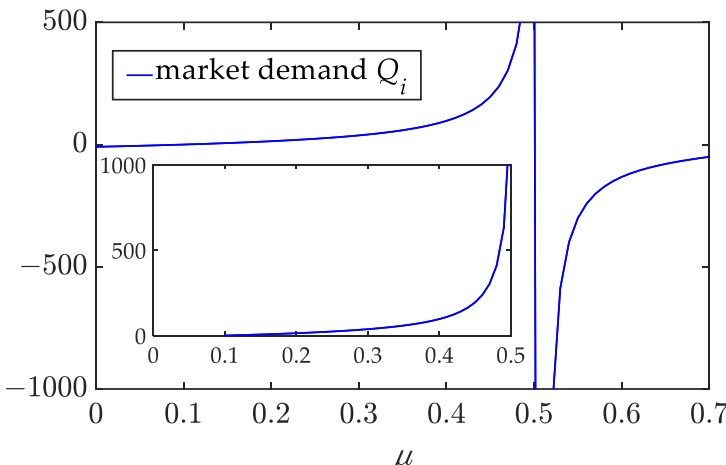

**Figure 15.** Effect of competition coefficient on the market demand.

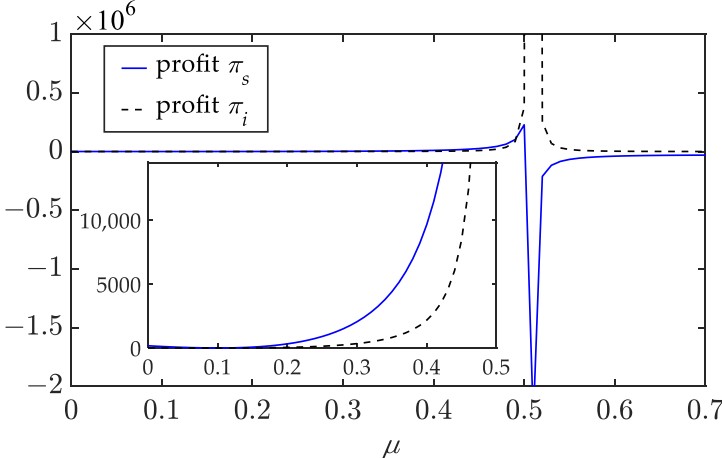

**Figure 16.** Effects of competition coefficient on the profits.

The figures reveal that when the level of competition is low, (1) both the shipping price and the brand value of the shipping company increase with $\mu$; (2) the freight service price, the effort levels, and market demands of the freight forwarders also increase with $\mu$; and (3) as the competition intensity increases, the profits of the shipping company and

the freight forwarders gradually increase. These findings allow us to derive the following insights.

When the degree of competition between the two freight forwarders is within a certain range, the more intense the competition is, the more efforts they will make to extend the brand value. These efforts can increase market demand, and their altruistic preferences can raise the shipping company's profit, leading to greater enthusiasm for brand building (i.e., the shipping company invests more in the brand building). As a result, the shipping company increases the shipping prices for the freight forwarders, and the latter increases the freight service prices to compensate their marketing expenses. The increase in shipping market demand ultimately leads to increased profits for parties and improves the overall profit of the supply chain. However, if the degree of competition is stronger, the above conclusion may not be valid.

## 6. Discussion and Conclusions

Following the behavioral economics, this research incorporated altruistic preferences into the cooperative mechanism of the SCSCS through a Stackelberg game, where the shipping company acts as the leader responsible for brand building and two freight forwarders act as the followers responsible for supplementary brand investment. The study presented the equilibrium decisions and profit functions of the various decision-making parties when the freight forwarders have altruistic preferences. Additionally, the research examined the impact of their altruistic preferences on the decisions and profits of the different parties involved in the SCSCS. The study's findings are summarized as follows.

(1)    When freight forwarders exhibit altruistic preferences and the coefficients of their altruistic preference are relatively low, it allows the shipping company to increase its brand value and the freight forwarders to engage in more supplementary branding efforts. When the parameters in the supply chain system meet certain conditions, freight forwarders' altruistic preferences can increase different parties' profits in the SCSCS, thereby increasing the overall profit of the supply chain. This suggests that the freight forwarders' altruistic behaviors can generate dual effects of altruism and self-interest.

(2)    The profits of the shipping company and the freight forwarders are positively correlated with the shippers' brand preferences and the freight forwarders' altruistic preferences under certain conditions.

The main contributions of this paper are as follows.

The first is theoretical contribution. This paper is one of only a few studies to provide such SCSCS game models. This paper obtains the optimal pricing of the shipping and freight service the brand value of shipping company and effort levels of freight Forwarders in the extending brand value and analyzes the advantages of the altruistic preference of forwarders, which further enriches the research content of the SCSCS.

The second is practical contribution. This paper can provide decision support models for SCSCS members to deal with pricing and the efforts of the extending brand value. Managers can make some operational decisions by changing the relevant parameters in the models. In a word, the models established in this paper can easily help managers optimize their own decisions and choose an appropriate altruistic intensity.

The examination of the equilibrium decisions provides the following practical suggestion for management.

(1)    When making altruistic decisions, freight forwarders should focus on increasing their efforts to extend the brand value in order to boost market demand and secure their own profits, rather than solely transferring profits to the shipping company.

(2)    The shipping company should increase its investment in brand building to maintain the stability of the supply chain structure, even though it experiences increased profit from being favored by freight forwarders' altruistic behaviors.

(3)  In a competing SCSCS, if one of the two freight forwarders exhibits altruistic behavior, it could facilitate the shipping company's brand-building efforts. However, the uncoordinated behavior could lead to intense competition between the freight forwarders, resulting in profit loss and destabilizing the SCSCS's structure. This implies that shipping companies should not only engage in moderate competition but also focus on reducing costs, innovating services, and coordinating development.

The limitations of this paper are as follows. First, the altruistic preference explored in this study represents only one type of behavioral preference among decision makers. Preferences such as overconfidence, mutual benefits, and fairness can significantly impact decision-making outcomes and the dynamics between competing stakeholders. Second, in reality, an SCSCS often has many freight forwarders, whereas we only considered two competing freight forwarders. Third, this research is not well integrated with new technologies. With the emergence of digital supply chain trends, the industry is transitioning towards a more interconnected and integrated network [45]. Technologies such as blockchain can play a transformative role in facilitating more informed and efficient decision-making processes [46,47]. Additionally, the integration of blockchain with other emerging technologies, such as the Internet of Things (IoT) and artificial intelligence (AI), can further enhance the decision-making capabilities of supply chain systems.

In summary, future research that incorporates various behavioral preferences and explores the potential of technologies such as blockchain can provide a more comprehensive and holistic understanding of decision making in supply chain systems. This knowledge can contribute to the development of more effective strategies and practices for managing supply chains, leading to improved profitability, stability, and overall performance of the system. In addition, the new block-chain technology [46,47] and digitization technology [45] can be connected with the green supply chain [48], and the safe operation and ship pollution prevention management system can be established and improved to achieve green and sustainable development [49] in future research on SCSCS.

**Supplementary Materials:** The following supporting information can be downloaded at: https://www.mdpi.com/article/10.3390/systems11080399/s1, S1: Real Case; S2: Proofs of lemmas and propositions; S3: Mathematical Derivation in Remark 1; S4: Range of the Altruistic Preference Degree; S5: Data Collection and Calculation Basis.

**Author Contributions:** Conceptualization, X.-Y.M. and D.-Q.S.; methodology, X.-Y.M., D.-Q.S. and S.-X.L.; software, X.-Y.M., D.-Q.S., Y.-T.L. and X.L.; validation, X.-Y.M. and D.-Q.S.; formal analysis, X.-Y.M. and D.-Q.S.; investigation, X.-Y.M. and D.-Q.S.; writing—original draft preparation, X.-Y.M. and D.-Q.S.; writing—review and editing, D.-Q.S. and S.-X.L.; visualization, D.-Q.S., Y.-T.L., H.-Q.M., L.-M.Z. and X.L.; supervision, D.-Q.S.; project administration, D.-Q.S. and S.-X.L.; funding acquisition, S.-X.L. and D.-Q.S. All authors have read and agreed to the published version of the manuscript.

**Funding:** This research was funded by the National Social Science Foundation of China, grant number 20BJL122, Hebei Key Research Institute of Humanities and Social Sciences at Universities, grant number HYZD202302, the Scientific Research Fund of Hebei Normal University of Science and Technology, grant number 2022YB028, and the National Natural Science Foundation of China, grant number 62006069.

**Informed Consent Statement:** Not applicable.

**Data Availability Statement:** Not applicable.

**Conflicts of Interest:** The authors declare no conflict of interest.

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
