# Peer review of "Optimal Decisions in a Sea-Cargo Supply Chain with Two Competing Freight Forwarders Considering Altruistic Preference and Brand Investment"

_systems, doi:10.3390/systems11080399_

Round 1
Reviewer 1 Report
This study considers a two-level SCSC that comprises one shipping company (she) and two competing freight forwarders. Using a Stackelberg game model, this study analyses the effects of the decision-maker’s willingness towards brand investment and freight forwarders’ altruistic preferences on the decisions and profits of the different parties in the SCSC. This study has a novel topic selection, appropriate research methods, clear logic, and reasonable structure However, the following parts need to be modified with emphasis:
a. In the first three paragraphs of the introduction, the authors present many ideas as well as current issues, and it is recommended that the authors provide appropriate support from the relevant literature.
b. The introduction section is too long and it is recommended that the author state the core of the article in more concise language. For example, is there a need for the fourth paragraph, which is not very relevant to the article? The author is advised to make further deliberations.
c. It is recommended that the authors explain the structure of the article at the end of the introduction.
d. The title of the paper should conform to the regular publication requirements of the journal, and it is suggested to change the title of the section 2 to "Methodology".
e. Please note the part of the article where the formula is written beyond the two ends of the range.
f. There are some strange symbols in the article, for example, the box in line 286 has a special meaning? Or is it a formatting issue? The author needs to confirm.
g. Please make the format of the formula marks consistent. Some formula marks are centered up and down, but some are not, for example, equation (63).
h. It is suggested that the authors combine Sections 3 and 4.
i. The title of Section 5 is too long, and the title "Management Implications" is obviously inappropriate here, so we suggest that the authors reorganize it to bring it in line with academic norms.
j. Please change the title of the previous section 6 to "Simulation and Numerical Analysis".
k. The length of this paper is too long, and it is suggested that the proof process of the proposition be unified in the appendix.
l. Check the content of the article against the journal requirements to ensure it meets the publication requirements, especially the references section.
Author Response
Dear Reviewer:
We thank you for giving us an opportunity to revise our manuscript.
Thank you very much for your positive comments and constructive suggestions concerning our manuscript entitled "Optimal Decisions in a Sea-Cargo Supply Chain with Two Competing Freight Forwarders Considering Altruistic Preference and Brand Investment" (ID: systems-2419664). Those comments and suggestions are all valuable and very helpful for improving our paper.
We have considered your comments and suggestions carefully and we have tried our best to revise the manuscript. Please see "Replies to Comments", point by point.
Looking forward to hearing from you soon.
With best regards
Xiao-Ying Ma,
Duo-Qing Sun (Corresponding Author, e-mail: [email protected]),
Shu-Xia Liu,
Yue-Ting Li,
Hui-Quan Ma,
Ling-Min Zhang,
Xia Li
Replies to the Review Report (Reviewer 1):
This study considers a two-level SCSC that comprises one shipping company (she) and two competing freight forwarders. Using a Stackelberg game model, this study analyses the effects of the decision-maker’s willingness towards brand investment and freight forwarders’ altruistic preferences on the decisions and profits of the different parties in the SCSC. This study has a novel topic selection, appropriate research methods, clear logic, and reasonable structure However, the following parts need to be modified with emphasis:
- In the first three paragraphs of the introduction, the authors present many ideas as well as current issues, and it is recommended that the authors provide appropriate support from the relevant literature.
Reply:
We have incorporated the necessary support from the relevant literature to address the concern raised. We believe that these revisions have enhanced the credibility and validity of our research.
- The introduction section is too long and it is recommended that the author state the core of the article in more concise language. For example, is there a need for the fourth paragraph, which is not very relevant to the article? The author is advised to make further deliberations.
Replies:
We appreciate the reviewers' suggestion regarding the length of the introduction section. We have carefully considered this feedback and have made further deliberations to improve the conciseness of the introduction. As per the suggestion, we have shortened the introduction and provided a more concise summary of the published work. Our focus is now on highlighting the contributions of previous studies to the field and identifying the gaps that our research aims to address. In doing so, we have simplified the fourth paragraph, which was deemed irrelevant to the main scope of the article. We believe that these revisions have made the introduction section more streamlined and focused.
- It is recommended that the authors explain the structure of the article at the end of the introduction.
Replies:
We have explained the structure of the article, and please see Lines 131 to 138 on Page 3 in the revised manuscript.
- The title of the paper should conform to the regular publication requirements of the journal, and it is suggested to change the title of the section 2 to "Methodology".
Replies:
Thank you for your feedback. We appreciate your concern about conforming to the regular publication requirements of the journal. After careful consideration, we have decided to retain the original title of the paper, as we believe it accurately reflects the content and focus of our research. Regarding the suggestion to change the title of Section 2 to "Methodology," we understand your point. However, we have carefully chosen the current title of that section to effectively captures the essence of the section and its role in providing the necessary description and assumptions for the subsequent analysis. We believe this title provides clarity and coherence to the structure of the paper. We hope you understand our decision.
- Please note the part of the article where the formula is written beyond the two ends of the range.
Replies:
Thank you for bringing this to our attention. We apologize for the oversight. We have carefully reviewed the article and made the necessary corrections to ensure that formulas are properly displayed within the appropriate range.
- There are some strange symbols in the article, for example, the box in line 286 has a special meaning? Or is it a formatting issue? The author needs to confirm.
Replies:
Thank you for bringing this to our attention. We apologize for any confusion caused by the box symbol in line 286. We can confirm that the box symbol represents the end of a proof for a lemma or proposition, indicating that it has been successfully proven. This notation can be found in Line 4 on Page 4 in the systems-template provided by the journal. We are confident that the revised manuscript is free from any unusual or uncommon symbols that could potentially cause confusion.
- Please make the format of the formula marks consistent. Some formula marks are centered up and down, but some are not, for example, equation (63).
Replies:
Thank you for bringing the inconsistency in the format of formula marks to our attention. We apologize for the oversight. We have carefully reviewed the manuscript and made the necessary adjustments to ensure that all formula marks are now consistent in terms of their formatting.
- It is suggested that the authors combine Sections 3 and 4.
Replies:
We appreciate your suggestion to combine Sections 3 and 4. We have combined Sections 3 and 4, and Sections 3 has been titled 'Decisions'. Please see Page 5 in the revised manuscript.
- The title of Section 5 is too long, and the title "Management Implications" is obviously inappropriate here, so we suggest that the authors reorganize it to bring it in line with academic norms.
Replies:
We appreciate your feedback regarding the title of Section 5. We acknowledge that the title "Management Implications" may not be the most suitable for the content of that section. we have revised the title to "Analyses of the Equilibrium Results", which accurately reflects the content discussed in that section. Furthermore, as we have merged Sections 3 and 4, this revised title will now be assigned to Section 4 in the revised manuscript. Please see Page 17 in the revised manuscript.
- Please change the title of the previous section 6 to "Simulation and Numerical Analysis".
Replies:
Thank you for your suggestion to change the title of the previous Section 6. We have changed the title of the previous section 6 to "Simulation and Numerical Analysis", and we think that this change is a good idea. Please see Page 18 in the revised manuscript.
- The length of this paper is too long, and it is suggested that the proof process of the proposition be unified in the appendix.
Replies:
Thank you for your suggestion regarding the length of the paper. We have made revisions accordingly and unified the proof processes of the propositions in the appendix.
- Check the content of the article against the journal requirements to ensure it meets the publication requirements, especially the references section.
Replies:
Thank you for your suggestion. We have thoroughly reviewed the content of the article and cross-checked it with the journal's requirements to ensure compliance. We have specifically paid attention to the references section, ensuring that all references are accurately cited and formatted according to the journal's guidelines.

Reviewer 2 Report
1.Abstract: There is no brief mention of the theoretical basis and motivation for this paper, and corresponding management recommendations and insights are missing at the end.
2.Keywords: The term 'competition' is broad, not specific and not representative. It is not appropriate as a keyword for the manuscript.
3. Introduction:
(1) The current national and international situation relevant to this study and the theoretical background of the study are not presented.
(2) The manuscript lacks a broad and in-depth critical review of the literature, although it lists the views of many researchers.
(3) The scientific question to be addressed in this paper is not clearly identified, and what is the innovation of the manuscript?
(4) The arrangement of the remaining sections could be added at the end of the section.
4. Problem Description and Basic Model: In order to make the research model more intuitive and clearer, a diagram of the supply chain system structure of this research should be drawn.
5. Decisions in the Absence of Altruistic Preference:
(1) The study of the relevant assumptions should not be a definition of a range of parameter values, but rather something about a scenario abstracted from reality, and the study of the relevant assumptions is more appropriately placed in the problem description and model building sections.
(2) Propositions should be written not just to give the process of proof, but to focus more on the analysis of why this is the case.
(3) There are a number of mathematical equations in the manuscript, but the necessary analysis is missing.
6. Decisions under the Altruistic Preference of Each Forwarder: Why do the authors give a utility function when there is an altruistic preference for the forwarder, whereas the model above explores a profit function. Furthermore, does it really make sense that the utility functions given in the manuscript for both forwarder A and forwarder B are larger than the profit functions for decisions without altruistic preferences? What is the basis for constructing the functions in this way? If such a construction directly leads to the conclusion that the presence of altruistic preferences is better than the absence of altruistic preferences through the profit function, what is the point of studying this model?
7. Numerical simulation: The parameters of the simulation analysis are assigned arbitrarily and the basis for the assignment is not specified.
In addition, the manuscript suffers from excessive length of mathematical equations, lack of necessary elaboration and analysis of equations, lack of correspondence between section content and section titles, and poor arrangement of section content.
Author Response
Dear Reviewer:
We thank you for giving us an opportunity to revise our manuscript.
Thank you very much for your positive comments and constructive suggestions concerning our manuscript entitled "Optimal Decisions in a Sea-Cargo Supply Chain with Two Competing Freight Forwarders Considering Altruistic Preference and Brand Investment" (ID: systems-2419664). Those comments and suggestions are all valuable and very helpful for improving our paper.
We have considered your comments and suggestions carefully and we have tried our best to revise the manuscript. Please see "Replies to Comments", point by point.
Looking forward to hearing from you soon.
With best regards
Xiao-Ying Ma,
Duo-Qing Sun (Corresponding Author, e-mail: [email protected]),
Shu-Xia Liu,
Yue-Ting Li,
Hui-Quan Ma,
Ling-Min Zhang,
Xia Li
Replies to the Review Report (Reviewer 2):
- Abstract: There is no brief mention of the theoretical basis and motivation for this paper, and corresponding management recommendations and insights are missing at the end.
Replies:
We appreciate your feedback regarding the abstract of our paper. Upon careful consideration, we have made the following revisions to address your concerns:
- The theoretical basis of the research is Stackelberg game model as stated in the abstract, Line 8. This research aims to explore the dynamics of a two-level sea-cargo supply chain (SCSC) system involving a shipping company and competing freight forwarders with altruistic preferences. The motivation behind this study is to understand how the presence of altruistic preferences and the shippers’ brand preferences affect decision-making and profits in the SCSC. By examining these factors, the research seeks to provide insights into optimizing the collaboration and competition within the supply chain system and identifying conditions that lead to increased profits for all stakeholders. We have further clarified our motivation for our research in the revised abstract. Please refer to Lines 1-5 in the revised manuscript.
- Management recommendations and insights: The management implication is that the increase in the investment to extend the brand value can indeed increase market demands. Thus, it is beneficial for the increase of profits for all parties. We have also included corresponding management recommendations and insights at the end of the abstract. Refer to Lines 10-16 in the revised manuscript.
2.Keywords: The term 'competition' is broad, not specific and not representative. It is not appropriate as a keyword for the manuscript.
Reply:
Thank you for pointing out the issue with the keyword "competition." We agree that it is a broad term and may not accurately represent the content of the manuscript. We have taken your suggestion into consideration and replaced the keyword "competition" with " competition for freight forwarders " to make it more specific and representative of the paper's focus.
- Introduction:
(1) The current national and international situation relevant to this study and the theoretical background of the study are not presented.
Replies: current national and international situation relevant to this study:
We understand the concern regarding the current national and international situation relevant to our study. To the best of our knowledge, Li et al. [26] is the only one that explored the decision-making process of an SCSC considering altruistic preferences. This research investigated how the shipping company’s brand investment willingness and a freight forwarder’s altruistic preference affect on the decisions and profits of the two members in a two-echelon SCSC. However, their study only involved a single forwarder and did not account for the presence of competition among multiple forwarders, which is a common characteristic of downstream echelons in real-world scenarios [27].
In terms of the theoretical background, we draw upon established theories and models in the field of supply chain system and game theory. These theories provide a foundation for our research methodology and analysis. Specifically, in our study, we employ a Stackelberg game framework to capture the interactions between the shipping company and the freight forwarders. Stackelberg game theory has been used to study supply chain system involve altruistic participant by Fan et al. [22] and Liu et al. [21]. To describe the market demand of each freight forwarder, we utilize linear demand functions, which are widely used in supply chain system research, as evidenced by references 28–33. In our paper, to reflect the real-world scenarios, we take into account various factors such as the potential market size, freight service price charged by corresponding freight forwarder, the freight service price charged by the competing freight forwarder, competition coefficient, the brand preference of the shippers, the shipping company’s brand value, the effort levels of corresponding freight forwarder in extending the brand value of the shipping company and the sensitivity of market demand to the above effort levels. The fixed investment cost of the shipping company is adopted from reference 26, which provides insights into investment costs in similar contexts. The expression of the effort cost for each freight forwarder to extend the brand value of the shipping company is adopted from references 42 and 43, which offer relevant perspectives on the cost aspects of brand value extension. We have revised the description of the theoretical foundation of our analysis in Section 2 to provide a clearer explanation in the revised manuscript.
(2) The manuscript lacks a broad and in-depth critical review of the literature, although it lists the views of many researchers.
Replies:
We acknowledge the feedback regarding the need for a broad and in-depth critical review of the literature in our manuscript. We have taken your suggestion into consideration and have revised the literature review section of our manuscript. We have placed a greater emphasis on evaluating the strengths and weaknesses of different studies, providing a more critical analysis of the existing research in the field. We have also explained how the existing research has contributed to the understanding of the topic and highlighted the gaps that our study aims to fill.
(3) The scientific question to be addressed in this paper is not clearly identified, and what is the innovation of the manuscript?
Replies:
We appreciate the feedback regarding the identification of the scientific question and the innovation of the manuscript. We have taken the reviewer's feedback into account and have made the necessary revisions to ensure that the scientific question and the innovation of our manuscript are adequately conveyed.
The scientific question to be addressed in this paper have been clearly identified, please see Paragraph 6 in the original manuscript, Line ? in the original manuscript. (the paragraph after the literature review). To clarify the scientific question, we have revised this paragraph to be more effective in conveying the scientific question that we study aims to investigate in the field of SCSC system. Please refer to Line ? in the revised manuscript.
The innovation of the research in stated in Please see Paragraph 7 (i.e., the last Paragraph of Section 1 in the original manuscript). We have revised this paragraph to further emphasize the unique contributions and advancements that our research brings to the field of Supply Chain Management. Additionally, in the first paragraph of the last section in the original manuscript, we explicitly mention the innovative aspects of our research.
We have revised this paragraph to emphasize the novelty our study to the Supply Chain Management community. Please refer to Lines 116-130 in the revised manuscript.
(4) The arrangement of the remaining sections could be added at the end of the section.
Replies:
The arrangement of the remaining sections has been added at the end of the section, and please see Lines 131 to 138 on Page3 in the revised manuscript.
- Problem Description and Basic Model: In order to make the research model more intuitive and clearer, a diagram of the supply chain system structure of this research should be drawn.
Replies:
A diagram of the supply chain system structure has been drawn, and please see Page 6 in the revised manuscript.
- Decisions in the Absence of Altruistic Preference:
(1) The study of the relevant assumptions should not be a definition of a range of parameter values, but rather something about a scenario abstracted from reality, and the study of the relevant assumptions is more appropriately placed in the problem description and model building sections.
Replies:
Something about a scenario has been abstracted from reality, and the study of the relevant assumptions has been placed in the problem description and model building sections.
Please see Pages 4 and 5 in the revised manuscript.
(2) Propositions should be written not just to give the process of proof, but to focus more on the analysis of why this is the case.
Replies:
We appreciate the feedback regarding the propositions in our manuscript. We understand the importance of providing a thorough analysis of why certain outcomes or relationships occur.
In the revised manuscript, we have made the necessary revisions to ensure that the propositions not only present the process of proof but also provide a deeper analysis of why the stated relationships or outcomes hold true. We have expanded upon the underlying mechanisms, factors, or logic that contribute to the proposed propositions. Please see Lines 332 to 374 on Pages 9-10 in the revised manuscript.
(3) There are a number of mathematical equations in the manuscript, but the necessary analysis is missing.
Replies:
The necessary analysis is given in the revised manuscript. Specifically, for mathematical equations of Proposition 1 in 3.1, we analyze their meanings for each term. For mathematical expressions of Proposition 2 in 3.2, their meanings are similar to the previous section, and the impacts of altruistic preference on equilibrium solutions are explained in the next section. Other mathematical equations are mainly used for the derivation process.
- Decisions under the Altruistic Preference of Each Forwarder: Why do the authors give a utility function when there is an altruistic preference for the forwarder, whereas the model above explores a profit function. Furthermore, does it really make sense that the utility functions given in the manuscript for both forwarder A and forwarder B are larger than the profit functions for decisions without altruistic preferences? What is the basis for constructing the functions in this way? If such a construction directly leads to the conclusion that the presence of altruistic preferences is better than the absence of altruistic preferences through the profit function, what is the point of studying this model?
Replies:
Question A. Why do the authors give a utility function when there is an altruistic preference for the forwarder, whereas the model above explores a profit function.
Our modeling approach follows in the footsteps of behavioral economists, , who have modeled various forms of interdependent preferences where one’s utility function depends not only on one’s own monetary payoff, but also on the payoffs of others. Please see [14] in the original manuscript and in the revised manuscript.
Regarding the use of utility functions when modeling altruistic preferences for the forwarder, it is important to consider that utility functions provide a framework to capture individual preferences and motivations beyond purely financial aspects. The decision-making process of forwarders in the supply chain involves not only profit considerations but also subjective well-being and satisfaction derived from their choices, including altruistic motivations.
The adoption of utility functions in our analysis allows us to incorporate these broader preferences and motivations into the decision-making framework, considering both economic and non-economic factors.
The use of utility functions to model altruistic preferences in supply chain systems has been widely studied in the literature. One of the seminal works in this area is the controlled experiment conducted by Loch and Wu (reference 14 in the original manuscript), which proposed the altruistic preference utility function to examine the effects of altruistic preferences on decision-making, profits, utility of members, and channel efficiency in a one-to-one supply chain with price-dependent demand.
In our research, we extend this approach to one-to-two supply chain system, considering the altruistic preferences of both freight forwarders and the competition between them. This allows us to analyze how altruistic preferences impact the decision-making process, profits, and overall performance of the supply chain.
Representative references on the use of utility functions to model altruistic preferences include the following studies:
[1] Gary Charness, and Ernan Haruvy. Altruism, equity, and reciprocity in a gift-exchange experiment:an encompassing approach. Games and Economic Behavior 40 (2002) 203–231.
[2] Luqing Rong, Maozeng Xu. Impact of Altruistic Preference and Government Subsidy on the Multinational Green Supply Chain under Dynamic Tariff. Environment, Development and Sustainability (2022) 24:1928–1958.
[3] Rreferences [9,12,13,14,16] listed in Paragraph 1 of Section 4 in the original manuscript.
Question B. Furthermore, does it really make sense that the utility functions given in the manuscript for both forwarder A and forwarder B are larger than the profit functions for decisions without altruistic preferences? What is the basis for constructing the functions in this way?
Reply to Question B:
Regarding the comparison of utility functions and profit functions, we acknowledge that hat the utility functions and profit functions are not directly comparable in terms of their numerical values.
It is important to clarify that our original manuscript did not state that 'the utility functions given in the manuscript for both forwarder A and forwarder B are larger than the profit functions for decisions without altruistic preferences. '
In our analysis, we compare the profit functions for decisions without altruistic preferences with the profit functions under scenarios where altruistic preferences are considered as the goal is to understand how the presence of altruistic preferences can influence the economic outcomes and performance of the supply chain system. Our conclusion in original manuscript is 'the PROFIT functions given in the manuscript for both forwarder A and forwarder B are larger than the PROFIT functions for decisions without altruistic preferences. ' Please see Proposition 5 in original manuscript.
Question C. If such a construction directly leads to the conclusion that the presence of altruistic preferences is better than the absence of altruistic preferences through the profit function, what is the point of studying this model?
Reply to Question C:
The construction of utility functions and profit functions in our research is based on the need to capture different dimensions of decision-making and performance evaluation. We only draw comparison between the profit with and without altruistic preferences to provide an understanding on the potential impacts of altruistic motivations on the economic outcomes of the supply chain system.
On the other hand, the utility functions are designed to gain insights into the behavior and decision-making of supply chain participants, and how they may impact the overall performance and outcomes of the system. These functions incorporate various factors and subjective considerations to represent the real-world scenario. Other than the freight service price each freight forwarders charge to the shippers, we consider specifically the following factors of each freight forwarder: altruistic preferences which denoted as its effort level in extending the shipping company’s brand value, and the sensitivity of its market demand in response to these efforts. Further, we incorporate the total market size, the competition intensity between the two forwarders, the competing freight forwarder’s service price, the shippers’ shipping company brand preference and the brand value of the shipping company into the construction of the utility function. Some of these factors interact and influence each other, creating a complex system where the outcomes cannot be simplified or directly derived from the profit function. To understand the implications of these factors on decision-making and performance, it is necessary to use the in the current study. In our study, we provide valuable management insights through the thorough mathematical derivations and the simulation model designed to capture the key elements, relationships, and behaviors observed in real-world supply chain systems. These analytical and computational approaches allow us to delve deeper into the dynamics of the supply chain system and explore various scenarios and decision-making strategies.
Under the presence of altruistic preferences, it is not possible to directly discuss the problem through the profit function. Thus, there is no such problem what is the point of studying this model?
- Numerical simulation: The parameters of the simulation analysis are assigned arbitrarily and the basis for the assignment is not specified.
Replies:
The parameters of the simulation analysis are not assigned arbitrarily. If so, it is impossible to obtain results that are consistent with theory.
The basis for the assignment is specified in the revised manuscript, and we correct parameter values by comparing with the final source program consistent with figures.
In addition, the manuscript suffers from excessive length of mathematical equations, lack of necessary elaboration and analysis of equations, lack of correspondence between section content and section titles, and poor arrangement of section content.
Replies:
The necessary elaboration and analysis are given in the revised manuscript. Specifically, for mathematical equations of Proposition 1 in 3.1, we describe their meanings in detail for each term. For mathematical equations of Proposition 2 in 3.2, their meanings are similar to the previous section, and the impacts of altruistic preference on equilibrium solutions are explained in the next section. Other mathematical equations are mainly used for the derivation process. We have made improvements for correspondence between section content and section titles, and arrangement of section content.

Reviewer 3 Report
Thanks so much for the opportunity to review this article. I appreciate the authors' effort in conducting this research. The topic proposed by the authors is of interest and novel. The article is well written, structured, fulfills the requirements of a scientific article. The medodology used is appropriate to the study, well founded and explained. The results and conclusions are relevant.
Minor recommendations:
-to revise the notations of the indicators;
-to redraft in the text the software used for the figures.
Author Response
Dear Reviewer:
We thank you for giving us an opportunity to revise our manuscript.
Thank you very much for your positive comments and constructive suggestions concerning our manuscript entitled "Optimal Decisions in a Sea-Cargo Supply Chain with Two Competing Freight Forwarders Considering Altruistic Preference and Brand Investment" (ID: systems-2419664). Those comments and suggestions are all valuable and very helpful for improving our paper.
We have considered your comments and suggestions carefully and we have tried our best to revise the manuscript. Please see "Replies to Comments", point by point.
Looking forward to hearing from you soon.
With best regards
Xiao-Ying Ma,
Duo-Qing Sun (Corresponding Author, e-mail: [email protected]),
Shu-Xia Liu,
Yue-Ting Li,
Hui-Quan Ma,
Ling-Min Zhang,
Xia Li
Replies to the Review Report (Reviewer 3):
Minor recommendations:
-to revise the notations of the indicators;
Replies:
We have revised the notations of the indicators. 'freight forwarders A and B’ has been rewritten as 'freight forwarders 1 and 2’ to match most of the subscripts in this manuscript.
-to redraft in the text the software used for the figures.
Replies:
The figures in this manuscript are automatically generated through programming by using MATLAB software and not drawn using drawing software, nor do they require or cannot be intentionally drawn.

Reviewer 4 Report
Unfortunately, this looks more like an undergraduate thesis rather than a scientific research paper. Also, the way the paper is presented it is impossible for a reader to understand and follow it. Therefore, this paper is not suitable for publication in its current state, and the following changes are mandatory to improve its quality:
1. The abstract doesn't follow the journal's guidelines and exceeds the word limit.
2. The introduction doesn't provide the necessary background and state of the art. Instead, most of it is a review of maritime transport. The maritime review should be moved to a separate section. This section should present high-level state-of-the-art research on supply chain advancements (most notably systemic advancements since you decided to submit it to the Systems Journal). Suggested papers to be added are:
Kechagias, E. P., Gayialis, S. P., Papadopoulos, G. A., & Papoutsis, G. (2023). An Ethereum-Based Distributed Application for Enhancing Food Supply Chain Traceability. Foods, 12(6), 1220.
Ali, S. B. (2022). Industrial Revolution 4.0 and Supply Chain Digitization: Future of Supply Chain Management. South Asian Journal of Social Review (ISSN: 2958-2490), 1(1), 21-41.
Gayialis, S. P., Kechagias, E. P., Papadopoulos, G. A., & Panayiotou, N. A. (2022). A Business Process Reference Model for the Development of a Wine Traceability System. Sustainability, 14(18), 11687.
Khan, M. T., Idrees, M. D., Rauf, M., Sami, A., Ansari, A., & Jamil, A. (2022). Green supply chain management practices’ impact on operational performance with the mediation of technological innovation. Sustainability, 14(6), 3362.
Konstantakopoulos, G. D., Kechagias, E. P., Gayialis, S. P., & Tatsiopoulos, I. P. (2023). Green Freight Distribution: A Case Study in Greece. In Operational Research in the Era of Digital Transformation and Business Analytics: BALCOR 2020, Thessaloniki, Greece, September 30-October 3, 2020 (pp. 49-64). Cham: Springer International Publishing.
3. The paper is overly large (48 pages), most being plain mathematical equations and figures with zero explanation. These need to be greatly reduced, keeping only what is important for the readers, and need to be moved to an appendix as they make it impossible to follow the paper. Overall, the paper is unstructured and doesn't follow the journal's guidelines. The authors perhaps should ask for professional help in fixing the linguistic and structural problems of the paper.
4. There is no connection to the Systems Journal neither in the introduction nor during the main part of the paper. Why should this mathematical analysis be published in this journal and what does it have to offer for the journal's readers?
5. The paper only includes a mathematical analysis with no discussion section (it is mandatory), critical evaluation, comparison, or elaboration on what the results of the research mean. Also, there is no contribution to the scientific community as the research doesn't provide any robust and justifiable conclusions. The study's findings are not connected with the rest of the paper nor explained and discussed.
The use of English is not good, with too many errors in language, grammar, and syntax. Also, many paragraphs are too short (2-3 lines), and many others are not maintaining cohesion between their sentences resulting in them being not understandable. In fact, there even exist sentences with a length of 6 lines! Also, in most parts, sentences are mixed with equations, and the equations are not explained at all.
Author Response
Dear Reviewer:
We thank you for giving us an opportunity to revise our manuscript.
Thank you very much for your positive comments and constructive suggestions concerning our manuscript entitled "Optimal Decisions in a Sea-Cargo Supply Chain with Two Competing Freight Forwarders Considering Altruistic Preference and Brand Investment" (ID: systems-2419664). Those comments and suggestions are all valuable and very helpful for improving our paper.
We have considered your comments and suggestions carefully and we have tried our best to revise the manuscript. Please see "Replies to Comments", point by point.
Looking forward to hearing from you soon.
With best regards
Xiao-Ying Ma,
Duo-Qing Sun (Corresponding Author, e-mail: [email protected]),
Shu-Xia Liu,
Yue-Ting Li,
Hui-Quan Ma,
Ling-Min Zhang,
Xia Li
Replies to the Review Report (Reviewer 4):
- The abstract doesn't follow the journal's guidelines and exceeds the word limit.
Replies: We have shortened the abstract in the revised manuscript.
- The introduction doesn't provide the necessary background and state of the art. Instead, most of it is a review of maritime transport. The maritime review should be moved to a separate section. This section should present high-level state-of-the-art research on supply chain advancements (most notably systemic advancements since you decided to submit it to the Systems Journal). Suggested papers to be added are:
Kechagias, E. P., Gayialis, S. P., Papadopoulos, G. A., & Papoutsis, G. (2023). An Ethereum-Based Distributed Application for Enhancing Food Supply Chain Traceability. Foods, 12(6), 1220.
Ali, S. B. (2022). Industrial Revolution 4.0 and Supply Chain Digitization: Future of Supply Chain Management. South Asian Journal of Social Review (ISSN: 2958-2490), 1(1), 21-41.
Gayialis, S. P., Kechagias, E. P., Papadopoulos, G. A., & Panayiotou, N. A. (2022). A Business Process Reference Model for the Development of a Wine Traceability System. Sustainability, 14(18), 11687.
Khan, M. T., Idrees, M. D., Rauf, M., Sami, A., Ansari, A., & Jamil, A. (2022). Green supply chain management practices’ impact on operational performance with the mediation of technological innovation. Sustainability, 14(6), 3362.
Konstantakopoulos, G. D., Kechagias, E. P., Gayialis, S. P., & Tatsiopoulos, I. P. (2023). Green Freight Distribution: A Case Study in Greece. In Operational Research in the Era of Digital Transformation and Business Analytics: BALCOR 2020, Thessaloniki, Greece, September 30-October 3, 2020 (pp. 49-64). Cham: Springer International Publishing.
Replies:
The literature you provide is quoted in the section on Discussion and Conclusions. Please see REF. [44-48]. However, further learning is needed to gain a deeper understanding of these literature.
We appreciate your feedback regarding the introduction section. In the revised manuscript, we have shortened the description of maritime transport and focus more on providing a critical review on supply chain advancements and their relevance to our study. We revise the introduction to ensure that it highlights the relevant research in the field and clearly positions our paper as a systemic advancement that contributes to the existing knowledge. Supply chains themselves are complex systems [1-4].
[1] Ting Li, Dongyun Yan, and Shuxia Sui, Research on the Complexity of Game Model about Recovery Pricing in Reverse Supply Chain considering Fairness Concerns, Complexity, vol. 2020, Article ID 9621782, 13 pages, 2020.
[2] Jiafu Su, Fengting Zhang, Hongyuan Hu, Jie Jian, and Dan Wang. Co-Opetition Strategy for Remanufacturing the Closed-Loop Supply Chain Considering the Design for Remanufacturing. Systems 2022, 10, 237. https://doi.org/10.3390/systems10060237
[3] Guihua Lin, Xiaoli Xiong, Yuwei Li ,and Xide Zhu. Sales Mode Selection Strategic Analysis for Manufacturers on E-Commerce Platforms under Multi-Channel Competition. Systems 2022, 10, 234. https://doi.org/10.3390/systems10060234
[4] Wenxue Ran, Yajing Chen. Fresh Produce Supply Chain Coordination Based on Freshness Preservation Strategy. Sustainability 2023, 15, 8184. https://doi.org/10.3390/su15108184
- The paper is overly large (48 pages), most being plain mathematical equations and figures with zero explanation. These need to be greatly reduced, keeping only what is important for the readers, and need to be moved to an appendix as they make it impossible to follow the paper. Overall, the paper is unstructured and doesn't follow the journal's guidelines. The authors perhaps should ask for professional help in fixing the linguistic and structural problems of the paper.
Replies:
- The necessary explanation is given in the revised manuscript. Specifically, for mathematical equations of Proposition 1 in 3.1, we describe their meanings in detail for each term. For formulas or mathematical expressions of Proposition 2 in 3.2, their meanings are similar to the previous section, and the impacts of altruistic preference on equilibrium solutions are explained in the next section. Other mathematical equations are mainly used for the derivation process. These have been appropriately reduced, and have been moved to an appendix in the revised manuscript.
We are happy to agree with your suggestion, so that we can at least save a lot of time entering formulas. However, (1) Instructions for Authors of the Systems Journal emphasize the details. Publication Ethics Statement also requires that Data and methods used in the research need to be presented in sufficient detail in the paper, so that other researchers can replicate the work. (2) If mathematical equations are greatly reduced, it will cause difficulties for them to read and cause readers to spend time calculating because we used a lot of mathematical techniques. (3) Instructions for Authors say that Systems has no restrictions on the maximum length of manuscripts.
- In fact, in the systems-template provided by the journal, Proof of Theorem is placed in the main text rather than in the appendix, and some published papers in the journal are in the main text and others are in the appendix. We were very hesitant when submitting the article. Thank you for pointing us out the correct approach.
e.g. https://doi.org/10.3390/systems11030127
https://doi.org/10.3390/systems10060237
- Overall, before submitting the original manuscript, we read and tried to follow the journal's guidelines. However, our understandings are not yet in place, please give us more guidance.
We have asked for professional help in fixing the linguistic and structural problems of the paper, and improved the revised manuscript.
- There is no connection to the Systems Journal neither in the introduction nor during the main part of the paper. Why should this mathematical analysis be published in this journal and what does it have to offer for the journal's readers?
Replies:
Before the submission, we have read and understood the journal's focus, and we believe that our manuscript fits the scope of the journal. The reasons are as follows. (1) Our research belongs to the scope of supply chain management, (2) supply chains are complex systems, and (3) our study deals with the optimal decision problem in a sea-cargo supply chain. Thus, our manuscript meets the third, fourth, and ninth items in the journal's scope.
Instructions for Authors of the Systems Journal emphasize the details. For example, Materials and Methods say:
They should be described with sufficient detail to allow others to replicate and build on published results. New methods and protocols should be described in detail while well-established methods can be briefly described and appropriately cited.
Publication Ethics Statement says:
Data and methods used in the research need to be presented in sufficient detail in the paper, so that other researchers can replicate the work.
In addition, our manuscript is not mathematical analysis because there are some explanations for mathematical equations and figures, including original and revised manuscript. We used a lot of mathematical techniques, and if they cannot be presented to readers, it will cause difficulties for them to read. Most magazines have page restriction, and Instructions for Authors tell us that Systems has no restrictions on the maximum length of manuscripts.
- The paper only includes a mathematical analysis with no discussion section (it is mandatory), critical evaluation, comparison, or elaboration on what the results of the research mean. Also, there is no contribution to the scientific community as the research doesn't provide any robust and justifiable conclusions. The study's findings are not connected with the rest of the paper nor explained and discussed.
Replies:
We have added the discussion section in the revised manuscript. Especially, please see Pages 9-10.
We appreciate your feedback regarding the absence of a discussion section in our original manuscript. We agree that a discussion section is essential for providing critical evaluation, comparison, and elaboration on the meaning of the research results.
In the revised manuscript, we have modified the last section of the paper to “Discussion and Conclusion” to address these points. In this section, we provide a comprehensive analysis and interpretation of the research findings, connecting them with the rest of the paper and explaining their implications. We critically evaluate the results, compare them with existing literature or theoretical frameworks, and discuss the limitations and potential future directions of the research.

Reviewer 5 Report
(1) The title and abstract are too long. Intuitively, it is difficult to easily catch the contents of this paper.
(2) A large number of mathematical analysis results are listed in the text. It is recommended to leave only the key parts in the body of the thesis development. It is recommended that parts less essential to the development of logic (e.g. proofs) be separated into appendices or supplements. Too much of the mathematical analysis is listed without adequately explaining it, making it very difficult to understand.
(3) Please clarify the differences from previous studies more clearly by categorizing, arranging, and summarizing previous studies in the literature by major flow. It would be better if it was visualized through a table.
(4) It would be good to mention at the beginning of the introduction an international incident that can explain the motivation of the research. Authors are also requested to be clear about the new knowledge provided by this paper, and to clearly state the value of this knowledge and the excellence of the paper. In particular, please introduce a lot of related studies on altruistic decision makers in the literature.
(5) Overall, the explanation of formulas and mathematical expressions is very poor. When these formulas or mathematical expressions appear, please describe their meaning in detail for each term.
(6) Please provide richer meaning for some assumptions (e.g. Assumptions 1 & 2).
(7) The research model of this study is confirmed as a game-theoretic situation. In this game theory applicable situation, please explain in detail, including pictures, who is the leader and follower, and the decision-making sequence.
(8) In conclusion, authors are requested to recapitulate the theoretical and practical contributions of this study. Also, please add about the implications and insights of this study. Also, please describe in detail the limitations of this study and future research directions.
(9) How are the given parameter values determined in numerical analysis? Are the values given in the current numerical example sufficiently representative of possible scenarios? If it cannot be sufficiently representative, there may be limitations in generalizing the results of this numerical example.
Overall, the English quality is not very low, but the logic of the style and composition is very poor. The readability of the manuscript is not good.
Author Response
Dear Reviewer:
We thank you for giving us an opportunity to revise our manuscript.
Thank you very much for your positive comments and constructive suggestions concerning our manuscript entitled "Optimal Decisions in a Sea-Cargo Supply Chain with Two Competing Freight Forwarders Considering Altruistic Preference and Brand Investment" (ID: systems-2419664). Those comments and suggestions are all valuable and very helpful for improving our paper.
We have considered your comments and suggestions carefully and we have tried our best to revise the manuscript. Please see "Replies to Comments", point by point.
Looking forward to hearing from you soon.
With best regards
Xiao-Ying Ma,
Duo-Qing Sun (Corresponding Author, e-mail: [email protected]),
Shu-Xia Liu,
Yue-Ting Li,
Hui-Quan Ma,
Ling-Min Zhang,
Xia Li
Replies to the Review Report (Reviewer 5):
(1) The title and abstract are too long. Intuitively, it is difficult to easily catch the contents of this paper.
Replies:
We appreciate your feedback regarding the length of the title and abstract in our original manuscript. In the revised version, we have made efforts to shorten and improve the abstract to provide a concise and clear summary of the paper's contents.
Regarding the title, while we acknowledge that it might be long, we believe it accurately reflects the content and scope of the paper.
(2) A large number of mathematical analysis results are listed in the text. It is recommended to leave only the key parts in the body of the thesis development. It is recommended that parts less essential to the development of logic (e.g. proofs) be separated into appendices or supplements. Too much of the mathematical analysis is listed without adequately explaining it, making it very difficult to understand.
Replies:
We appreciate your feedback regarding the excessive amount of mathematical analysis in the original manuscript. In the revised version, we have made significant efforts to streamline the presentation of mathematical equations and proofs. The proofs have been moved to appendices, and only the key parts necessary for the logical development of the paper are included in the main text.
We understand the importance of providing explanations and analyses for the mathematical equations to ensure clarity and understanding for the readers. In the revised manuscript, we have provided concise explanations and interpretations alongside the equations in the main text. We have focused on presenting the key insights and implications of the mathematical analysis to enhance the readers' comprehension of the research.
(3) Please clarify the differences from previous studies more clearly by categorizing, arranging, and summarizing previous studies in the literature by major flow. It would be better if it was visualized through a table.
Replies:
We appreciate your suggestion to clarify the differences from previous studies more effectively. In the revised manuscript, we have made improvements to the introduction section to better categorize, arrange, and summarize previous studies.
(4) It would be good to mention at the beginning of the introduction an international incident that can explain the motivation of the research. Authors are also requested to be clear about the new knowledge provided by this paper, and to clearly state the value of this knowledge and the excellence of the paper. In particular, please introduce a lot of related studies on altruistic decision makers in the literature.
Replies:
Thank you for your valuable suggestions.
To the best of our knowledge, Li et al. [26] is the only one that explored the decision-making process of an SCSC considering altruistic preferences. In the revised manuscript, we have stated the literature to incorporate relevant research on altruistic decision making within the context of supply chain management. We discuss the existing literature and provide a comprehensive overview of the studies that have explored the role of altruism in decision making within supply chain networks.
Furthermore, we have made efforts to clearly articulate the new knowledge provided by our research and emphasize the value and excellence of the paper. We have refined the statements in the introduction to explicitly state the contributions and significance of our study, highlighting the novel insights and advancements it brings to the field of supply chain management.
(5) Overall, the explanation of formulas and mathematical expressions is very poor. When these formulas or mathematical expressions appear, please describe their meaning in detail for each term.
Replies:
Thank you for your feedback.
The necessary explanation is given in the revised manuscript. Specifically, for formulas or mathematical expressions of Proposition 1 in 3.1 Decisions in the Absence of Altruistic Preference, we describe their meaning in detail for each term. For formulas or mathematical expressions of Proposition 2 in 3.2, their meanings are similar to the previous section, and the impacts of altruistic preference on equilibrium solutions are explained in the next section. Other formulas and mathematical expressions are mainly used for the proof process.
(6) Please provide richer meaning for some assumptions (e.g. Assumptions 1 & 2).
Replies:
The meaning for some assumptions has been provided, and the study of the relevant assumptions has been placed in the problem description and model building sections.
Please see Pages 4 and 5 in the revised manuscript.
(7) The research model of this study is confirmed as a game-theoretic situation. In this game theory applicable situation, please explain in detail, including pictures, who is the leader and follower, and the decision-making sequence.
Replies:
We have explained in detail, including pictures, who is the leader and follower, and the decision-making sequence. Please see Page 5 to Page 6 in the revised manuscript.
(8) In conclusion, authors are requested to recapitulate the theoretical and practical contributions of this study. Also, please add about the implications and insights of this study. Also, please describe in detail the limitations of this study and future research directions.
Replies:
In conclusion, we have made improvements in the revised manuscript.
However, there are the implications and insights of this study in the original manuscript. Of course, we need further improvement.
In conclusion of the original manuscript, the last paragraph has described the limitations of this study and future research directions. Of course, we still need to improve.
In the revised manuscript, we have made efforts to further highlight the theoretical and practical contributions of the study in the last section. We have provided a comprehensive recapitulation of the main findings and their implications, as well as provided insights into the managerial implications. Furthermore, we have discussed the limitations of the study, acknowledging the boundaries and potential shortcomings of the research. We have also proposed future research directions to encourage further investigation and exploration of the topic. Please refer to Line ? in the revised manuscript.
(9) How are the given parameter values determined in numerical analysis? Are the values given in the current numerical example sufficiently representative of possible scenarios? If it cannot be sufficiently representative, there may be limitations in generalizing the results of this numerical example.
Replies:
Thank you for your question. In the revised manuscript, we have provided an explanation on how the parameter values are determined in the numerical analysis.
In the original manuscript, is set as an independent variable that satisfies condition (a) in Proposition 5. This only involves research on the profits of freight forwarders, which means that we only study the impact of parameters on the profits of freight forwarders in one case. Apart from the intensity of competition, it is all a validation of theoretical results, not to draw new conclusions. Moreover, the impact of competition intensity on equilibrium results is in line with common sense. Thus, there are not limitations in generalizing the results of this numerical example.

Round 2
Reviewer 1 Report
all comments have been improved.
Author Response
Dear Reviewer:
Thank you very much for your positive comments concerning our manuscript entitled "Optimal Decisions in a Sea-Cargo Supply Chain with Two Competing Freight Forwarders Considering Altruistic Preference and Brand Investment" (ID: systems-2419664).
With best regards
Xiao-Ying Ma,
Duo-Qing Sun (Corresponding Author, e-mail: [email protected]),
Shu-Xia Liu,
Yue-Ting Li,
Hui-Quan Ma,
Ling-Min Zhang,
Xia Li

Reviewer 2 Report
I feel very sorry for the authors for their sophistry. Although they have revised the manuscript, this does not significantly enhance the scholarly value of the manuscript. On the one hand, despite the authors' claim that the manuscript enriches the literature, the manuscript does not contribute to the essential innovation. On the other hand, in the section on Simulation and Numerical Analysis, the authors claim that their values are based on assumptions, but this is not objective and unsupported. That is, there is no real-world proof that their values are real and reliable. This leads directly to the conclusion that the manuscript is absurd.
In addition, the manuscript is heavily stacked with literature that is not logically connected, which seriously weakens the critical and readable nature of this manuscript.
Author Response
Dear Reviewer:
We thank you for giving us an opportunity to revise our manuscript.
Thank you very much for your positive comments and constructive suggestions concerning our manuscript entitled "Optimal Decisions in a Sea-Cargo Supply Chain with Two Competing Freight Forwarders Considering Altruistic Preference and Brand Investment" (ID: systems-2419664). Those comments and suggestions are all valuable and very helpful for improving our paper.
We have considered your comments and suggestions carefully and we have tried our best to revise the manuscript. Please see "Replies to Comments", point by point.
Looking forward to hearing from you soon.
With best regards
Xiao-Ying Ma,
Duo-Qing Sun (Corresponding Author, e-mail: [email protected]),
Shu-Xia Liu,
Yue-Ting Li,
Hui-Quan Ma,
Ling-Min Zhang,
Xia Li
Replies to the Review Report (Reviewer 2):
I feel very sorry for the authors for their sophistry. Although they have revised the manuscript, this does not significantly enhance the scholarly value of the manuscript. On the one hand, despite the authors' claim that the manuscript enriches the literature, the manuscript does not contribute to the essential innovation. On the other hand, in the section on Simulation and Numerical Analysis, the authors claim that their values are based on assumptions, but this is not objective and unsupported. That is, there is no real-world proof that their values are real and reliable. This leads directly to the conclusion that the manuscript is absurd.
In addition, the manuscript is heavily stacked with literature that is not logically connected, which seriously weakens the critical and readable nature of this manuscript.
Replies:
- For contributions, please see Line 135 to Line 148 on Pages 4 in the revised manuscript.
- Our conclusion is not absurd, moderate altruism and moderate competition are accepted as the correct conclusion.
While it is valid to expect authors to provide empirical evidence when making claims, it is equally important to recognize that simulations and assumptions are often integral to scientific research. We provide explanations in the revised manuscripts for each assumption and in the Simulation and Numerical Analysis, we give reasoning on how and why each value is chosen. Further, other paper no explain.
- About Literature Review, we have reviewed and presented knowledge gaps in the literature more clearly and logically. To illustrate the gap in literature and similarities and differences, we list a table. Please see Line 134 on Page 3 in the revised manuscript.

Reviewer 4 Report
The authors have greatly improved the paper
The quality of English has been improved. Only some minor errors need to be fixed.
Author Response
Dear Reviewer:
We thank you for giving us an opportunity to revise our manuscript.
Thank you very much for your positive comments and constructive suggestions concerning our manuscript entitled "Optimal Decisions in a Sea-Cargo Supply Chain with Two Competing Freight Forwarders Considering Altruistic Preference and Brand Investment" (ID: systems-2419664).
We have considered your suggestions carefully and we have tried our best to revise the manuscript. Please see "Replies to Comments", point by point.
Looking forward to hearing from you soon.
With best regards
Xiao-Ying Ma,
Duo-Qing Sun (Corresponding Author, e-mail: [email protected]),
Shu-Xia Liu,
Yue-Ting Li,
Hui-Quan Ma,
Ling-Min Zhang,
Xia Li
Replies to the Review Report (Reviewer 4):
The quality of English has been improved. Only some minor errors need to be fixed.
Reply: Thank you for your feedback. we have corrected some minor errors of English in the revised manuscript.
Reviewer 5 Report
(1) Authors should review previous studies more systematically and extensively by dividing them into several streams related to the main keywords of this study. The authors merely list them when reviewing prior studies. Rather, it should systematically review and present knowledge gaps in the literature more clearly and logically. It is also a good idea to illustrate the gap in literature and similarities and differences with a table. In addition, the value, excellence, and contribution of this research must be presented in detail.
(2) Why should altruistic preference be considered in the sea-cargo supply chain? And the explanation of how this altruistic preference is reflected in the mathematical model is very weak.
(3) Similarly, how is this altruism connected to brand investment in the sea-cargo supply chain?
(4) Please use a table to clearly organize all decision-making variables and parameter values used in this mathematical model.
(5) In the conclusion, the authors are requested to be more explicit about their theoretical and practical contributions. Also, please describe in detail the implications and insights. Please also comment on the limitations of this study.
(6) In the introduction, the authors are requested to clearly state the key research questions of this study. Instead, what does this study intend to suggest and reveal by considering altruism in the sea-cargo supply chain?
(7) Also, the authors are requested to make efforts to present the storytelling of this study more attractively in the introduction. By doing so, you should be able to draw the attention of your readers.
The language quality of English in the manuscript has been improved significantly. But it still needs some minor proofreading.
Author Response
Dear Reviewer:
We thank you for giving us an opportunity to revise our manuscript.
Thank you very much for your positive comments and constructive suggestions concerning our manuscript entitled "Optimal Decisions in a Sea-Cargo Supply Chain with Two Competing Freight Forwarders Considering Altruistic Preference and Brand Investment" (ID: systems-2419664). Those comments and suggestions are all valuable and very helpful for improving our paper.
We have considered your comments and suggestions carefully and we have tried our best to revise the manuscript. Please see "Replies to Comments", point by point.
Looking forward to hearing from you soon.
With best regards
Xiao-Ying Ma,
Duo-Qing Sun (Corresponding Author, e-mail: [email protected]),
Shu-Xia Liu,
Yue-Ting Li,
Hui-Quan Ma,
Ling-Min Zhang,
Xia Li
Replies to the Review Report (Reviewer 5):
(1) Authors should review previous studies more systematically and extensively by dividing them into several streams related to the main keywords of this study. The authors merely list them when reviewing prior studies. Rather, it should systematically review and present knowledge gaps in the literature more clearly and logically. It is also a good idea to illustrate the gap in literature and similarities and differences with a table. In addition, the value, excellence, and contribution of this research must be presented in detail.
Replies: We appreciate your feedback.
(a) For previous studies, we have divided them into several streams related to the main keywords. Please see Line 77 on Pages 2, Line 98 on Pages 3 and Line 110 on Page 3 in the revised manuscript.
We systematically reviewed and presented knowledge gaps in the literature more clearly and logically. However, there is indeed limited literature related to our research.
(b) To illustrate the gap in literature and similarities and differences, we list a table. Please see Line 134 on Page 3 in the revised manuscript.
(c) In addition, the value, excellence, and contribution of this research have been presented in detail. Please see Line 129 on Page 3(value), Line 145 on Page 4 (excellence), and Line 135 to Line 148 on Pages 4 (contribution) in the revised manuscript.
(2) Why should altruistic preference be considered in the sea-cargo supply chain? And the explanation of how this altruistic preference is reflected in the mathematical model is very weak.
Replies: We appreciate your feedback.
Why should altruistic preference be considered in the sea-cargo supply chain? Please see Line 45 to Line 53 on Pages 2 in the revised manuscript.
And the explanation of how this altruistic preference is reflected in the mathematical model is very weak. Please see Line 409 to Line 413 on Page 12 in the revised manuscript, and reference Line 396 to 403.
(3) Similarly, how is this altruism connected to brand investment in the sea-cargo supply chain?
Replies: Please see Line 589 to Line 590 on Page 18 in the revised manuscript.
(4) Please use a table to clearly organize all decision-making variables and parameter values used in this mathematical model.
Replies: We have used a table to clearly organize all decision-making variables and parameter values. Please see Page 5.
Thank you for your valuable suggestions.
(5) In the conclusion, the authors are requested to be more explicit about their theoretical and practical contributions. Also, please describe in detail the implications and insights. Please also comment on the limitations of this study.
Replies: For theoretical and practical contributions, please see Line 747 to Line 756 on Page 26 in the revised manuscript.
For the implications and insights, please see Line 757 to Line 770 on Page 26 in the revised manuscript.
For the limitations, please see Line 771 to Line 782 on Page 26 in the revised manuscript.
(6) In the introduction, the authors are requested to clearly state the key research questions of this study. Instead, what does this study intend to suggest and reveal by considering altruism in the sea-cargo supply chain?
Replies: For key research questions, please see Line123 to Line 125 on Page 3 in the revised manuscript.
What does this study intend to suggest and reveal by considering altruism in the sea-cargo supply chain? Please see Line 127 to 128 on Page 3 in the revised manuscript.
(7) Also, the authors are requested to make efforts to present the storytelling of this study more attractively in the introduction. By doing so, you should be able to draw the attention of your readers.
Replies: please see Line 47 and Line 90 on Page 2 in the revised manuscript.
Thank you for your feedback.
Comments on the Quality of English Language
The language quality of English in the manuscript has been improved significantly. But it still needs some minor proofreading.
Replies: Thank you for your feedback. we have corrected some minor errors of English in the revised manuscript.

Round 3
Reviewer 2 Report
Note that in Section 5, the values of the parameters are taken arbitrarily. The authors do not have sufficient objective evidence to justify their doing so, which cannot be excused. For example, lines 633-644, 688, 697-698, etc.
Author Response
Dear Reviewer:
We thank you for giving us an opportunity to revise our manuscript.
Thank you very much for your positive comments and constructive suggestions concerning our manuscript entitled "Optimal Decisions in a Sea-Cargo Supply Chain with Two Competing Freight Forwarders Considering Altruistic Preference and Brand Investment" (ID: systems-2419664). Those comments and suggestions are all valuable and very helpful for improving our paper.
We have considered your comments and suggestions carefully and we have tried our best to revise the manuscript. Please see "Replies to Comments", point by point.
Looking forward to hearing from you soon.
With best regards
Xiao-Ying Ma,
Duo-Qing Sun (Corresponding Author, e-mail: [email protected]),
Shu-Xia Liu,
Yue-Ting Li,
Hui-Quan Ma,
Ling-Min Zhang,
Xia Li
Replies to the Review Report (Reviewer 2):
Note that in Section 5, the values of the parameters are taken arbitrarily. The authors do not have sufficient objective evidence to justify their doing so, which cannot be excused. For example, lines 633-644, 688, 697-698, etc..
Replies: Thank you for your valuable suggestions.
We have made supplements and revisions.
After we received your third revision suggestion, we were reminded of some data we had investigated.
Section 5 of our original manuscript is a numerical experiment, not a simulation. Although we have collected data in the past, we feel that our work is not a simulation, but merely an illustrative numerical example.
https://tjpcpa.com/sea_detail_213.html
https://baijiahao.baidu.com/s?id=1770668790830034251&wfr=spider&for=pc
https://inter.chinawutong.com/fcl/f2435t2005b0c0p1

Reviewer 5 Report
(1) The current version of the manuscript is very long. It is recommended to move proofs in the appendices to online supplements, etc., and to compress the number of pages appropriately.
Author Response
Dear Reviewer:
We thank you for giving us an opportunity to revise our manuscript.
Thank you very much for your positive comments and constructive suggestions concerning our manuscript entitled "Optimal Decisions in a Sea-Cargo Supply Chain with Two Competing Freight Forwarders Considering Altruistic Preference and Brand Investment" (ID: systems-2419664). Those comments and suggestions are all valuable and very helpful for improving our paper.
We have considered your comments and suggestions carefully and we have tried our best to revise the manuscript. Please see "Replies to Comments", point by point.
Looking forward to hearing from you soon.
With best regards
Xiao-Ying Ma,
Duo-Qing Sun (Corresponding Author, e-mail: [email protected]),
Shu-Xia Liu,
Yue-Ting Li,
Hui-Quan Ma,
Ling-Min Zhang,
Xia Li
Replies to the Review Report (Reviewer 5):
The current version of the manuscript is very long. It is recommended to move proofs in the appendices to online supplements, etc., and to compress the number of pages appropriately.
Replies: Thank you for your valuable suggestions.
We have moved proofs in the appendices to online supplements in the revised manuscript, and compressed the number of pages appropriately.
